# Transformers Learn Temporal Difference Methods for In-Context Reinforcement Learning

**Jiuqi Wang** [* 1]   **Ethan Blaser** [* 1]   **Hadi Daneshmand** [2]   **Shangtong Zhang** [1]

## Abstract

In-context learning refers to the learning ability of a model during inference time without adapting its parameters. The input (i.e., prompt) to the model (e.g., transformers) consists of both a context (i.e., instance-label pairs) and a query instance. The model is then able to output a label for the query instance according to the context during inference. A possible explanation for in-context learning is that the forward pass of (linear) transformers implements iterations of gradient descent on the instance-label pairs in the context. In this paper, we prove by construction that transformers can also implement temporal difference (TD) learning in the forward pass, a phenomenon we refer to as in-context TD. We demonstrate the emergence of in-context TD after training the transformer with a multi-task TD algorithm, accompanied by theoretical analysis. Furthermore, we prove that transformers are expressive enough to implement many other policy evaluation algorithms in the forward pass, including residual gradient, TD with eligibility trace, and average-reward TD.

## 1. Introduction

In-context learning has emerged as one of the most remarkable abilities of large language models (Brown et al., 2020; Lieber et al., 2021; Rae et al., 2021; Black et al., 2022). In in-context learning, the input (i.e., prompt) to the model consists of both a context (i.e., instance-label pairs) and a query instance. The model then outputs a label for the query instance during inference (i.e., the forward pass). An

*Equal contribution. The order is determined by tossing a fair coin. [1]Department of Computer Science, University of Virginia, Charlottesville, the United States [2]MIT LIDS/Boston University, Boston, the United States. Correspondence to: Shangtong Zhang <shangtong@virginia.edu>.

*Proceedings of the 1ˢᵗ Workshop on In-Context Learning at the 41ˢᵗ International Conference on Machine Learning*, Vienna, Austria. 2024. Copyright 2024 by the author(s).

example of the model input and output could be

$$\underbrace{5 \to \text{number}; a \to \text{letter}; 6 \to}_{\text{input}} \underbrace{\text{number}}_{\text{output}}, \qquad (1)$$

where "$5 \to$ number; $a \to$ letter" is the context consisting of two instance-label pairs and "$6$" is the query instance. Based on the context, the model (e.g., Team et al. (2023); Touvron et al. (2023); Achiam et al. (2023)) infers the label "number" for the query "$6$". Remarkably, this entire process occurs during the model's inference time without any adjustment to the model's parameters. Understanding the mechanism behind in-context learning has recently garnered significant attention (Garg et al., 2022; Akyürek et al., 2023; von Oswald et al., 2023; Ahn et al., 2024).

The example in (1) illustrates a supervised learning problem. In the canonical machine learning framework (Bishop, 2006), this supervised learning problem is typically solved by first training a classifier based on the instance-label pairs in the context using methods such as gradient descent, and then asking the classifier to predict the label for the query instance. Remarkably, Akyürek et al. (2023); von Oswald et al. (2023); Ahn et al. (2024) show that transformers are able to implement this gradient descent training process in their forward pass without adapting any of their parameters, providing a possible explanation for in-context learning.

Beyond supervised learning, intelligence involves sequential decision-making, where Reinforcement Learning (RL, Sutton & Barto (2018)) has emerged as a successful paradigm. Can transformers preform in-context RL during inference, and how? To address these questions, we start with a simple evaluation problem in a Markov Reward Process (MRP, Puterman (2014)). In an MRP, an agent transitions from state to state at every time step. We denote the sequence of states that the agent visits by $(S_0, S_1, S_2, \dots)$. At each state, the agent receives a reward. We denote the sequence of rewards that the agent receives along the way as $(r(S_0), r(S_1), r(S_2), \dots)$. The evaluation problem is to estimate the value function $v$, which computes for each state the expected total (discounted) rewards the agent will receive in the future. An example of the desired input-output

could be

$$\underbrace{S_0 \to r(S_0); S_1 \to r(S_1); S_2 \to r(S_2); s}_{\text{input}} \to \underbrace{v(s)}_{\text{output}}. \quad (2)$$

Remarkably, the above task is fundamentally different from supervised learning as the goal is to predict the value $v(s)$ and not the immediate reward $r(s)$. Moreover, the query state $s$ is arbitrary and does not have to be $S_3$. Temporal Difference learning (TD, Sutton (1988)) is the most widely used RL algorithm for solving such evaluation problems in (2). And it is well known that TD is *not* gradient descent (Sutton & Barto, 2018).

In this work, we make three main contributions. **First**, we prove by construction that transformers are expressive enough to implement TD in the forward pass, a phenomenon we refer to as *in-context TD*. In other words, transformers can solve problem (2) during inference time via in-context TD. Beyond the most straightforward TD, transformers can also implement many other policy evaluation algorithms, including residual gradient (Baird, 1995), TD with eligibility trace (Sutton, 1988), and average-reward TD (Tsitsiklis & Roy, 1999). In particular, to implement average-reward TD, transformers require the use of multi-head attention and over-parameterized prompts, e.g.,

$$\underbrace{S_0 \to r(S_0)\,\square; S_1 \to r(S_1)\,\square; S_2 \to r(S_2)\,\square; s}_{\text{input}} \to \underbrace{v(s)}_{\text{output}}.$$

Here, "$\square$" acts as a dummy placeholder that the transformers will use as "memory" during inference. **Second**, we empirically demonstrate that by training transformers with TD on multiple randomly generated evaluation problems, in-context TD emerges. In other words, the learned transformer parameters closely match our construction in proofs. We call this training scheme *multi-task TD*. **Third**, we bridge the gap between our theories and empirical results by showing that for a single layer transformer, the transformer parameters required in the proof to implement in-context TD is in a subset of the invariant set of the training algorithm multi-task TD.

## 2. Background

**Transformers and Linear Self-Attention.** All vectors in this paper are column vectors. We denote the identity matrix in $\mathbb{R}^n$ by $I_n$ and an $m \times n$ all-zero matrix by $0_{m \times n}$. We use $Z^\top$ to denote transpose of $Z$ and use both $\langle x, y \rangle$ and $x^\top y$ to denote the inner product. Given a prompt $Z \in \mathbb{R}^{d \times n}$, standard single-head self-attention (Vaswani et al., 2017) processes the prompt by $\text{Attn}_{W_k, W_q, W_v}(Z) \doteq W_v Z \,\text{softmax}\big(Z^\top W_k^\top W_q Z\big)$, where $W_v \in \mathbb{R}^{d \times d}, W_k \in \mathbb{R}^{m \times d}$, and $W_q \in \mathbb{R}^{m \times d}$ represent the value, key and query

weight matrices, respectively. The softmax function is applied to each row. Linear attention has recently drawn more attention (Schlag et al., 2021; von Oswald et al., 2023; Ahn et al., 2024), where the softmax function is replaced by an identity function. Given a prompt $Z \in \mathbb{R}^{(2d+1) \times (n+1)}$, we follow Ahn et al. (2024) and define linear self-attention as

$$\text{LinAttn}(Z; P, Q) \doteq PZM(Z^\top QZ), \quad (3)$$

where $P \in \mathbb{R}^{(2d+1) \times (2d+1)}$ and $Q \in \mathbb{R}^{(2d+1) \times (2d+1)}$ are parameters and $M \in \mathbb{R}^{(n+1) \times (n+1)}$ is a *fixed* mask of the input matrix $Z$, defined as

$$M \doteq \begin{bmatrix} I_n & 0_{n \times 1} \\ 0_{1 \times n} & 0 \end{bmatrix}. \quad (4)$$

Note that we can view $P$ and $Q$ as reparameterizations of the original weight matrices for simplifying presentation. The mask $M$ is introduced for in-context learning, following Ahn et al. (2024), to designate the last column of $Z$ as the query and the first $n$ columns as the context. We use this fixed mask in most of this work. However, the linear self-attention mechanism can be altered using a different mask $M'$, when necessary, by defining $\text{LinAttn}(Z; P, Q, M') = PZM'(Z^\top QZ)$. In an $L$-layer transformer with parameters $\{(P_l, Q_l)\}_{l=0,\ldots,L-1}$, the input $Z_0$ evolves layer by layer as

$$
\begin{aligned}
Z_{l+1} &\doteq Z_l + \frac{1}{n}\text{LinAttn}_{P_l, Q_l}(Z_l) \\
&= Z_l + \frac{1}{n} P_l Z_l M(Z_l^\top Q_l Z_l). \quad (5)
\end{aligned}
$$

Here $\frac{1}{n}$ is a normalization factor simplifying presentation. We follow the convention in von Oswald et al. (2023); Ahn et al. (2024) and use

$$
\begin{aligned}
&\text{TF}_L(Z_0; \{P_l, Q_l\}_{l=0,1,\ldots L-1}) \\
&\doteq -Z_L[2d+1, n+1] \quad (6)
\end{aligned}
$$

to denote the output of the $L$-layer transformer, given an input $Z_0$. Note that $Z_l[2d+1, n+1]$ is the bottom-right element of $Z_l$.

**In-Context Supervised Learning as Gradient Descent.** A linear regression task can be represented by an instance distribution $d_\mathcal{X}$ and a ground truth weight $w_*$. A training set $\{(x^{(i)} \in \mathbb{R}^{2d}, y^{(i)} \in \mathbb{R})\}_{i=1,\ldots,n}$ is usually constructed by sampling $n$ instances $\{x^{(i)}\}$ from $d_\mathcal{X}$ in an i.i.d. manner and constructing the targets as $y^{(i)} \doteq w_*^\top x^{(i)}$. For a new instance $x^{(n+1)}$ sampled from $d_\mathcal{X}$, the goal is to predict the correct target $y^{(n+1)}$. To demonstrate in-context learning, one constructs a prompt matrix as $Z_0 \doteq \begin{bmatrix} x^{(1)} & \cdots & x^{(n)} & x^{(n+1)} \\ y^{(1)} & \cdots & y^{(n)} & 0 \end{bmatrix}$, where the bottom right zero reflects that the target for $x^{(n+1)}$ is unknown. The $L$-layer transformer is trained via gradient descent to minimize

the following in-context loss

$$\mathbb{E}_{(d_{\mathcal{X}}, w_*) \sim d_{\text{task}}, Z_0 \sim d_{\mathcal{X}}}$$
$$\left[ (\text{TF}_L(Z_0; \{P_l, Q_l\}_{l=0}^{L-1}) - w_*^\top x^{(n+1)})^2 \right], \qquad (7)$$

where we have assumed that there is a distribution $d_{\text{task}}$ over such regression tasks. When a new regression task $(d_{\mathcal{X}}^{\text{test}}, w_*^{\text{test}})$ is sampled from $d_{\text{task}}$ and a new input $Z_0^{\text{test}}$ is constructed, the trained transformer, using $Z_0^{\text{test}}$ as input, approximates the target $\langle x^{(n+1),\text{test}}, w_*^{\text{test}} \rangle$. This is a form of meta-learning (Vilalta & Drissi, 2002). Surprisingly, the transformer's ability to achieve this stems from its implementation of gradient descent *within* its forward pass. As proved by (Ahn et al., 2024), by minimizing the in-context loss in (7), we may end up with a transformer parameterized by, say $\{(P_l^*, Q_l^*)\}_{l=0,\ldots,L-1}$, that has the following remarkable effect. Feeding the prompt $Z_0$ into this $L$-layer transformer, we get $Z_1, \ldots, Z_L$ following (5). We denote the right bottom element of $Z_l$ as $y_l^{(n+1)}$. (Ahn et al., 2024) then prove that for $l = 0, 1, \ldots, L$, we have $y_l^{(n+1)} = -w_l^\top x^{(n+1)}$, where $w_{l+1} \doteq w_l + \frac{1}{n} \sum_{i=1}^n (y^{(i)} - w_l^\top x^{(i)}) x^{(i)}$ with $w_0 = 0$. This sequence $\{w_l\}$ mirrors that produced by running gradient descent on the demonstrations $\{(x^{(i)}, y^{(i)})\}$ to minimize the squared loss $\frac{1}{n} \sum_{i=1}^n (y^{(i)} - w^\top x^{(i)})^2$. In other words, unrolling this transformer layer by layer is equivalent to performing gradient descent iteration by iteration.

**Reinforcement Learning.** We consider an infinite horizon Markov Decision Process (MDP, Puterman (2014)) with a finite state space $\mathcal{S}$, a finite action space $\mathcal{A}$, a reward function $r_{\text{MDP}} : \mathcal{S} \times \mathcal{A} \to \mathbb{R}$, a transition function $p_{\text{MDP}} : \mathcal{S} \times \mathcal{S} \times \mathcal{A} \to [0, 1]$, a discount factor $\gamma \in [0, 1)$, and an initial distribution $p_0 : \mathcal{S} \to [0, 1]$. An initial state $S_0$ is sampled from $p_0$. At a time $t$, an agent at a state $S_t$ takes an action $A_t \sim \pi(\cdot|S_t)$, where $\pi : \mathcal{A} \times \mathcal{S} \to [0, 1]$ is the policy being followed by the agent, receives a reward $R_{t+1} \doteq r_{\text{MDP}}(S_t, A_t)$, and transitions to a successor state $S_{t+1} \sim p_{\text{MDP}}(\cdot|S_t, A_t)$. If the policy $\pi$ is fixed, the MDP can be simplified to a Markov Reward Process (MRP) where transitions and rewards are determined solely by the current state: $S_{t+1} \sim p(\cdot|S_t)$ with $R_{t+1} \doteq r(S_t)$. Here $p(s'|s) \doteq \sum_a \pi(a|s) p_{\text{MDP}}(s'|s, a)$ and $r(s) \doteq \sum_a \pi(a|s) r_{\text{MDP}}(s, a)$. In this work, we consider the policy evaluation problem where the policy $\pi$ is fixed. So it suffices to consider only an MRP represented by the tuple $(p_0, p, r)$, and trajectories $(S_0, R_1, S_1, R_2, \ldots)$ sampled from it. The value function of this MRP is defined as $v(s) \doteq \mathbb{E}\left[ \sum_{i=t+1}^\infty \gamma^{i-t-1} R_i | S_t = s \right]$. Estimating the value function $v$ is one of the fundamental tasks in RL. To this end, one can consider a linear architecture. Let $\phi : \mathcal{S} \to \mathbb{R}^d$ be the feature function. The goal is then to find a weight vector $w \in \mathbb{R}^d$ such that for each $s$, the estimated value $\hat{v}(s; w) \doteq w^\top \phi(s)$ approximates $v(s)$. TD is a preva-

lent method for learning this weight vector, which updates $w$ iteratively as

$$w_{t+1}$$
$$= w_t + \alpha_t \big( R_{t+1} + \gamma \hat{v}(S_{t+1}; w_t) - \hat{v}(S_t; w_t) \big) \nabla \hat{v}(S_t; w_t)$$

$$= w_t + \alpha_t \left( R_{t+1} + \gamma w_t^\top \phi(S_{t+1}) - w_t^\top \phi(S_t) \right) \phi(S_t), \quad (8)$$

where $\{\alpha_t\}$ is a sequence of learning rates. Notably, TD is not a gradient descent algorithm. It is instead considered as a *semi-gradient* algorithm because the gradient is only taken with respect to $\hat{v}(S_t; w_t)$ and does not include the dependence on $\hat{v}(S_{t+1}; w_t)$ (Sutton & Barto, 2018). Including this dependency modifies the update to

$$w_{t+1} = w_t + \alpha_t \left( R_{t+1} + \gamma w_t^\top \phi(S_{t+1}) - w_t^\top \phi(S_t) \right)$$
$$\cdot (\phi(S_t) - \gamma \phi(S_{t+1})), \qquad (9)$$

known as the (naive version of) residual gradient method (Baird, 1995).[1] The update in (8) is also called TD(0) – a special case of the TD($\lambda$) algorithm (Sutton, 1988). TD($\lambda$) employs an eligibility trace that accumulates the gradients as $e_{-1} \doteq 0$, $e_t \doteq \gamma \lambda e_{t-1} + \phi(S_t)$ and updates $w$ iteratively as

$$w_{t+1} = w_t + \alpha_t (R_{t+1} + \gamma w_t^\top \phi(S_{t+1}) - w_t^\top \phi(S_t)) e_t.$$

The hyperparameter $\lambda$ controls the decay rate of the trace. If $\lambda = 0$, we recover (8). On the other end with $\lambda = 1$, it is known that TD($\lambda$) recovers Monte Carlo (Sutton, 1988). Another important setting in RL is the average-reward setting (Puterman, 2014; Sutton & Barto, 2018), focusing on the rate of receiving rewards, without using a discount factor $\gamma$. The average reward $\bar{r}$ is defined as $\bar{r} \doteq \lim_{T \to \infty} \frac{1}{T} \sum_{t=1}^T \mathbb{E}[R_t]$. Similar to the value function in the discounted setting, a differential value function $\bar{v}(s)$ is defined for the average-reward setting as $\bar{v}(s) \doteq \mathbb{E}\left[ \sum_{i=t+1}^\infty (R_i - \bar{r})|S_t = s \right]$. One can similarly estimate $\bar{v}(s)$ using a linear architecture with a vector $w$ as $w^\top \phi(s)$. Average-reward TD (Tsitsiklis & Roy, 1999) updates $w$ iteratively as

$$w_{t+1} = w_t + \alpha_t (R_{t+1} - \bar{r}_{t+1}$$
$$+ w_t^\top \phi(S_{t+1}) - w_t^\top \phi(S_t)) \phi(S_t),$$

where $\bar{r}_t \doteq \frac{1}{t} \sum_{i=1}^t R_i$ is the empirical average of the received reward.

## 3. Transformers Can Implement In-Context TD(0)

In this section, we prove that transformers are expressive enough to implement TD(0) in its forward pass. Given a

---

[1] This is a naive version because the update does not account for the double sampling issue. We refer the reader to Chapter 11 of Sutton & Barto (2018) for detailed discussion.

trajectory $(S_0, R_1, S_1, R_2, S_3, R_4, \ldots, S_n)$ sampled from an MRP, using as shorthand $\phi_i \doteq \phi(S_i)$, we define for $l = 0, 1, \ldots, L - 1$

$$Z_0 = \begin{bmatrix} \phi_0 & \ldots & \phi_{n-1} & \phi_n \\ \gamma\phi_1 & \ldots & \gamma\phi_n & 0 \\ R_1 & \ldots & R_n & 0 \end{bmatrix},$$

$$P_l^{\text{TD}} \doteq \begin{bmatrix} 0_{2d \times 2d} & 0_{2d \times 1} \\ 0_{1 \times 2d} & 1 \end{bmatrix}, \quad (10)$$

$$Q_l^{\text{TD}} \doteq \begin{bmatrix} -C_l^\top & C_l^\top & 0_{d \times 1} \\ 0_{d \times d} & 0_{d \times d} & 0_{d \times 1} \\ 0_{1 \times d} & 0_{1 \times d} & 0 \end{bmatrix}.$$

Here $Z_0 \in \mathbb{R}^{(2d+1) \times (n+1)}$ is the prompt matrix, $C_l \in \mathbb{R}^{d \times d}$ is an arbitrary matrix, and $\{(P_l^{\text{TD}}, Q_l^{\text{TD}})\}_{l=0,1,\ldots,L-1}$ are the parameters of the $L$-layer transformer. We then have

**Theorem 3.1** (Forward pass as TD(0)). *Consider the $L$-layer linear transformer following* (5), *using the mask* (4), *parameterized by* $\{P_l^{TD}, Q_l^{TD}\}_{l=0,\ldots,L-1}$ *in* (10). *Let* $y_l^{(n+1)}$ *be the bottom right element of the $l$-th layer's output, i.e.,* $y_l^{(n+1)} \doteq Z_l[2d+1, n+1]$. *Then, it holds that* $y_l^{(n+1)} = -\langle \phi_n, w_l \rangle$, *where* $\{w_l\}$ *is defined as* $w_0 = 0$ *and*

$$\begin{aligned} &w_{l+1} \\ =\ &w_l + \frac{1}{n} C_l \sum_{j=0}^{n-1} \left( R_{j+1} + \gamma w_l^\top \phi_{j+1} - w_l^\top \phi_j \right) \phi_j. \end{aligned} \quad (11)$$

The proof is in Appendix A.1 and with numerical verification in Appendix E as a sanity check. Notably, Theorem 3.1 holds for any $C_l$. In particular, if $C_l = \alpha_l I$, then the update (11) becomes a batch version of TD(0) in (8). For a general $C_l$, the update (11) can be regarded as preconditioned batch TD(0) (Yao & Liu, 2008). Theorem 3.1 precisely demonstrates that transformers are expressive enough to implement iterations of TD in its forward pass. We call this *in-context TD*. It should be noted that although the construction of $Z_0$ in (10) uses $\phi_n$ as the query state for conceptual clarity, any arbitrary state $s \in \mathcal{S}$ can serve as the query state and Theorem 3.1 still holds. In other words, by replacing $\phi_n$ with $\phi(s)$, the transformer will then estimate $v(s)$. Notably, if the transformer has only one layer, i.e., $L = 1$, there are other parameter configurations that can also implement in-context TD(0).

**Corollary 3.2.** *Consider the 1-layer linear transformer following* (5), *using the mask* (4). *Consider the following parameters*

$$P_0^{TD} \doteq \begin{bmatrix} 0_{2d \times 2d} & 0_{2d \times 1} \\ 0_{1 \times 2d} & 1 \end{bmatrix},$$

$$Q_0^{TD} \doteq \begin{bmatrix} -C_l^\top & 0_{d \times d} & 0_{d \times 1} \\ 0_{d \times d} & 0_{d \times d} & 0_{d \times 1} \\ 0_{1 \times d} & 0_{1 \times d} & 0 \end{bmatrix} \quad (12)$$

*Then, it holds that* $y_1^{(n+1)} = -\langle \phi_n, w_1 \rangle$, *where* $w_1$ *is defined as*

$$w_1 = w_0 + \frac{1}{n} C_l \sum_{j=0}^{n-1} \left( R_{j+1} + \gamma w_0^\top \phi_{j+1} - w_0^\top \phi_j \right) \phi_j$$

*with* $w_0 = 0$.

The proof is in Appendix A.2. An observant reader may notice that this corollary holds primarily because $w_0 = 0$, making it a unique result for $L = 1$. Nevertheless, this special case helps understand a few empirical and theoretical results below.

# 4. Transformers Do Implement In-Context TD(0)

It has been observed that in-context gradient descent emerges during the minimization of the in-context regression loss (7) via gradient descent. In this section, we demonstrate the emergence of in-context TD both theoretically and empirically.

**Multi-Task Temporal Difference Learning.** The in-context regression loss essentially trains the transformer with multiple regression tasks. Inspired by this, we propose to train the transformer with multiple evaluation tasks from multiple MRPs. Recall, an MRP is defined by the tuple $(p_0, p, r)$. For the evaluation problem, the feature function $\phi$ also matters. We therefore define an evaluation task to be the tuple $(p_0, p, r, \phi)$. Assuming a distribution $d_{\text{task}}$ over these tuples, we sample evaluation tasks from this distribution. For each sampled task, we apply TD to train the transformer to solve the corresponding evaluation problem, as described in the following multi-task TD algorithm (Algorithm 1).

Recall that $\text{TF}_L(Z_0; \theta)$ and $\text{TF}_L(Z_0'; \theta)$ are intended to estimate $v(S_{t+n+1})$ and $v(S_{t+n+2})$ respectively. So Algorithm 1 essentially applies TD using $(S_{t+n+1}, R_{t+n+2}, S_{t+n+2})$ to train the transformer. Ideally, when a new prompt $Z_{\text{test}}$ is constructed using a trajectory from a new evaluation task $(p_0, p, r, \phi)_{\text{test}} \sim d_{\text{task}}(\cdot)$, we would like the predicted value $\text{TF}_L(Z_{\text{test}}; \theta)$ with $\theta$ from Algorithm 1 to be close to the value of the query state in $Z_{\text{test}}$. This problem is a multi-task meta-learning problem, a well-explored area with many existing methodologies (Beck et al., 2023). However, the unique and significant aspect of our work is the demonstration that in-context TD emerges in the learned transformer, providing a novel *explanation* for how the model solves the problem.

## 4.1. Theoretical Analysis

The problem that Algorithm 1 aims to solve is highly non-convex and non-linear (the linear transformer is still a non-linear function). We analyze a simplified version of Algorithm 1 and leave the treatment to the full version for

**Algorithm 1** Multi-Task Temporal Difference Learning

1: **Input:** context length $n$, MRP sample length $\tau$, number of training MRPs $k$, learning rate $\alpha$, discount factor $\gamma$, transformer parameters $\theta \doteq \{P_l, Q_l\}_{l=0,1,\ldots L-1}$
2: **for** $i \leftarrow 1$ **to** $k$ **do**
3:     Sample $(p_0, p, r, \phi)$ from $d_{\text{task}}$   // see, e.g., Algorithm 2 in Appendix B
4:     Sample $(S_0, R_1, S_1, R_2, \ldots, S_\tau, R_{\tau+1}, S_{\tau+1})$ from the MRP $(p_0, p, r)$
5:     **for** $t = 0, \ldots, \tau - n - 1$ **do**
6:         $Z_0 \leftarrow \begin{bmatrix} \phi_t & \cdots & \phi_{t+n-1} & \phi_{t+n+1} \\ \gamma\phi_{t+1} & \cdots & \gamma\phi_{t+n} & 0 \\ R_{t+1} & \cdots & R_{t+n} & 0 \end{bmatrix}$
7:         $Z_0' \leftarrow \begin{bmatrix} \phi_{t+1} & \cdots & \phi_{t+n} & \phi_{t+n+2} \\ \gamma\phi_{t+2} & \cdots & \gamma\phi_{t+n+1} & 0 \\ R_{t+2} & \cdots & R_{t+n+1} & 0 \end{bmatrix}$
8:         $\theta \leftarrow \theta + \alpha(R_{t+n+2} + \gamma\text{TF}_L(Z_0'; \theta) - \text{TF}_L(Z_0; \theta))\nabla_\theta\text{TF}_L(Z_0; \theta)$   // TD
9:     **end for**
10: **end for**

future work. In particular, we study the single layer case with $L = 1$ and let $\theta \doteq (P_0, Q_0)$ be the parameters of the single-layer transformer. We consider expected updates, i.e.,

$$\theta_{k+1} = \theta_k + \alpha_k\Delta(\theta_k),$$

with

$$\Delta(\theta) \doteq \mathbb{E}\left[(R + \gamma\text{TF}_1(Z_0', \theta) - \text{TF}_1(Z_0, \theta))\nabla\text{TF}_1(Z_0, \theta)\right]. \tag{13}$$

Here the expectation integrates both the randomness in sampling $(p_0, p, r, \phi)$ from $d_{\text{task}}$ and the randomness in constructing $(R, Z_0, Z_0')$ thereafter. We sample $(S_0, R_1, S_1, \ldots, S_{n+1}, R_{n+2}, S_{n+2})$ following $(p_0, p, r)$ and construct using shorthand $\phi_i \doteq \phi(S_i)$

$$Z_0 \doteq \begin{bmatrix} \phi_0 & \cdots & \phi_{n-1} & \phi_{n+1} \\ \gamma\phi_1 & \cdots & \gamma\phi_n & 0 \\ R_1 & \cdots & R_n & 0 \end{bmatrix},$$
$$Z_0' \doteq \begin{bmatrix} \phi_1 & \cdots & \phi_n & \phi_{n+2} \\ \gamma\phi_2 & \cdots & \gamma\phi_{n+1} & 0 \\ R_2 & \cdots & R_{n+1} & 0 \end{bmatrix}, R \doteq R_{n+2}. \tag{14}$$

The structure of $Z_0$ and $Z_0'$ is similar to those in Algorithm 1. The main difference is that we do not use the sliding window. We recall that $(p_0, p, r, \phi)$ are random variables with joint distribution $d_{\text{task}}$. Here, $\phi$ is essentially a random matrix taking value in $\mathbb{R}^{d \times |\mathcal{S}|}$, represented as, $\phi = [\phi(s)]_{s \in \mathcal{S}}$. We use $\triangleq$ to denote "equal in distribution" and make the following assumptions.

**Assumption 4.1.** The random matrix $\phi$ is independent of $(p_0, p, r)$.

**Assumption 4.2.** $\Pi\phi \triangleq \phi, \Lambda\phi \triangleq \phi$, where $\Pi$ is any $d$-dimensional permutation matrix and $\Lambda$ is any diagonal matrix in $\mathbb{R}^d$ where each diagonal element of $\Lambda$ can only be $-1$ or 1.

Those assumptions are easy to satisfy. For example, as long as the elements of the random matrix $\phi$ are i.i.d. from a symmetric distribution centered at zero, e.g., a uniform distribution on $[-1, 1]$, then both assumptions hold. We say a set $\Theta$ is an invariant set of (13) if for any $k$, $\theta_k \in \Theta \implies \theta_{k+1} \in \Theta$. Define

$$\theta_*(\eta, c, c') \doteq$$
$$\left(P_0 = \begin{bmatrix} 0_{2d \times 2d} & 0_{2d \times 1} \\ 0_{1 \times 2d} & \eta \end{bmatrix}, Q_0 = \begin{bmatrix} cI_d & 0_{d \times d} & 0_{d \times 1} \\ c'I_d & 0_{d \times d} & 0_{d \times 1} \\ 0_{1 \times d} & 0_{1 \times d} & 0 \end{bmatrix}\right).$$

**Theorem 4.3.** *Let Assumptions 4.1 and 4.2 hold. For the* (14) *construction of* $(R, Z_0, Z_0')$, *then* $\Theta_* \doteq \{\theta_*(\eta, c, c')|\eta, c, c' \in \mathbb{R}\}$ *is an invariant set of* (13).

The proof is in Appendix A.3. Theorem 4.3 demonstrates that once $\theta_k$ enters $\Theta_*$ at some $k$, it can never leave, i.e., $\Theta_*$ is a candidate set that the update (13) can *possibly* converge to. Consider a subset $\Theta_*' \subset \Theta_*$ with a stricter constraint $c' = 0$, i.e., $\Theta_*' \doteq \{\theta_*(\eta, c, 0)|\eta, c \in \mathbb{R}\}$. Corollary 3.2 then confirms that all parameters in $\Theta_*'$ implement in-context TD.

That being said, whether (13) is guaranteed to converge to $\Theta_*$, or further to $\Theta_*'$, is left for future work.

### 4.2. Empirical Analysis

We now empirically study Algorithm 1. To this end, we construct $d_{\text{task}}$ based on Boyan's chain (Boyan, 1999), a canonical environment for diagnosing RL algorithms. We keep the structure of Boyan's chain but randomly generate initial distributions $p_0$, transition probabilities $p$, reward functions $r$, and the feature function $\phi$. Details of this random generation process are provided in Algorithm 2 with Figure 2 visualizing Boyan's chain, both in Appendix B.

For the linear transformer specified in (5), we first consider the autoregressive case following (Akyürek et al., 2023; von Oswald et al., 2023), where all the transformer layers share the same parameters, i.e., $P_l \equiv P_0$ and $Q_l \equiv Q_0$ for $l = 0, 1, \ldots, L-1$. We consider a three layer transformer ($L = 3$). Importantly, all elements of $P_0$ and $Q_0$ are equally trainable – we did not force any element of $P_0$ and $Q_0$ to be 0. We then run Algorithm 1 with Boyan's chain based evaluation tasks (i.e., $d_{\text{task}}$) to train this autoregressive transformer. The dimension of the feature is $d = 4$ (i.e., $\phi(s) \in \mathbb{R}^4$). Other hyperparameters of Algorithm 1 are specified in Appendix C.1.

Figure 1a visualizes the final learned $P_0$ and $Q_0$ by Al-

gorithm 1 after 4000 MRPs (i.e., $k = 4000$), which closely match our specifications $P^{\text{TD}}$ and $Q^{\text{TD}}$ in (10) with $C_l = I_d$. In Figure 1b, we visualize the element-wise learning progress of $P_0$ and $Q_0$. We observe that the bottom right element of $P_0$ increases (the $P_0[-1, -1]$ curve) while the average absolute value of all other elements remain close to zero (the "Avg Abs Others" curve), closely aligning with $P^{\text{TD}}$ up to some scaling factor. Furthermore, the trace of the upper left $d \times d$ block of $Q_0$ approaches $-d$ (the $\text{tr}(Q_0[:d, :d])$ curve), and the trace of the upper right block (excluding the last column) approaches $d$ (the $\text{tr}(Q_0[:d, d:2d])$ curve). Meanwhile, the average absolute value of all the other elements in $Q_0$ remain near zero, aligning with $Q^{\text{TD}}$ using $C_l = I_d$ up to some scaling factor.

More empirical analysis is provided in the Appendix. In particular, besides showing the parameter-wise convergence in Figure 1, we also use other metrics including value difference, implicit weight similarity, and sensitivity similarity, inspired by von Oswald et al. (2023); Akyürek et al. (2023), to examine the learned transformer. We also study **normal transformers without parameter sharing** (Appendix C.3), as well as **different choices of hyperparameters** in Algorithm 1. Furthermore, we empirically investigate the original **softmax-based transformers** (Appendix D). The overall conclusion is the same – in-context TD emerges in the transformers learned by Algorithm 1. Notably, Theorem 3.1 and Corollary 3.2 suggests that for $L = 1$, there are two distinct ways to implement in-context TD (i.e., (10) v.s. (12)). Our empirical results in Appendix C.2 show that Algorithm 1 ends up with (12) in Corollary 3.2 for $L = 1$, aligning well with Theorem 4.3. For $L = 2, 3, 4$, Algorithm 1 always ends up with (10) in Theorem 3.1, as shown in Figure 3 in Appendix C.2. We also empirically observed that for in-context TD to emerge, the task distribution $d_{\text{task}}$ has to be "difficult" enough. For example, if $(p_0, p)$ or $\phi$ are always fixed, we did not observe the emergence of in-context TD.

## 5. Transformers Can Implement More RL Algorithms

In this section, we prove that transformers are expressive enough to implement three additional well-known RL algorithms in the forward pass. We warm up with the (naive version of) residual gradient (RG). We then move to the more difficult TD($\lambda$). This section culminates with average-reward TD, which requires multi-head linear attention and memory within the prompt. We do note that whether those three RL algorithms will emerge after training is left for future work.

**Residual Gradient.** The construction of RG is an easy

extension of Theorem 3.1. We define

$$P_l^{\text{RG}} = P_l^{\text{TD}}, Q_l^{\text{RG}} \doteq \begin{bmatrix} -C_l^\top & C_l^\top & 0_{d \times 1} \\ C_l^\top & -C_l^\top & 0_{d \times 1} \\ 0_{1 \times d} & 0_{1 \times d} & 0 \end{bmatrix}$$
$$\in \mathbb{R}^{(2d+1) \times (2d+1)}. \quad (15)$$

**Corollary 5.1** (Forward pass as Residual Gradient). *Consider the L-layer linear transformer following* (5), *using the mask* (4), *parameterized by* $\left\{ P_l^{RG}, Q_l^{RG} \right\}_{l=0,\ldots,L-1}$ *in* (15). *Define* $y_l^{(n+1)} \doteq Z_l[2d+1, n+1]$. *Then, it holds that* $y_l^{(n+1)} = -\langle \phi_n, w_l \rangle$, *where* $\{w_l\}$ *is defined as* $w_0 = 0$ *and*

$$w_{l+1} = w_l + \frac{1}{n} C_l \sum_{j=0}^{n-1} \left( R_{j+1} + \gamma w_l^\top \phi_{j+1} - w_l^\top \phi_j \right)$$
$$\cdot (\phi_j - \gamma \phi_{j+1}). \quad (16)$$

The proof is in A.4 with numerical verification in Appendix E as a sanity check. Again, if $C_l \doteq \alpha_l I_d$, then (16) can be regarded as a batch version of (9). For a general $C_l$, it is then preconditioned batch RG. Notably, Figure 1 empirically demonstrates that Algorithm 1 eventually ends up with in-context TD instead of in-context RG. This matches the conventional wisdom in the RL community that TD is usually superior to the naive RG (see, e.g., Zhang et al. (2020) and references therein).

**TD($\lambda$).** Incorporating eligibility traces is an important extension of TD(0). We now demonstrate that by using a different mask, transformers are able to implement in-context TD($\lambda$). We define

$$M^{\text{TD}(\lambda)} \doteq \begin{bmatrix} 1 & 0 & 0 & 0 & \cdots & 0 & 0 \\ \lambda & 1 & 0 & 0 & \cdots & 0 & 0 \\ \vdots & \vdots & \vdots & \vdots & \ddots & \vdots & \vdots \\ \lambda^{n-1} & \lambda^{n-2} & \lambda^{n-3} & \lambda^{n-4} & \cdots & 1 & 0 \\ 0 & 0 & 0 & 0 & \cdots & 0 & 0 \end{bmatrix}$$
$$\in R^{(n+1) \times (n+1)}. \quad (17)$$

Notably, if $\lambda = 0$, the above mask for TD($\lambda$) recovers the mask for TD(0) in (4).

**Corollary 5.2** (Forward pass as TD($\lambda$)). *Consider the L-layer linear transformer parameterized by* $\left\{ P_l^{TD}, Q_l^{TD} \right\}_{l=0,\ldots,L-1}$ *as specified in* (10) *with the input mask used in* (5) *being* $M^{TD(\lambda)}$ *in* (17). *Define* $y_l^{(n+1)} \doteq Z_l[2d+1, n+1]$. *Then, it holds that* $y_l^{(n+1)} = -\langle \phi_n, w_l \rangle$ *where* $\{w_l\}$ *is defined with* $w_0 = 0, e_0 = 0, e_j = \lambda e_{j-1} + \phi_j$, *and*

$$w_{k+1} = w_k + \frac{1}{n} C_k \sum_{i=0}^{n-1} \left( r_{i+1} + \gamma w_k^\top \phi_{i+1} - w_k^\top \phi_i \right) e_i.$$

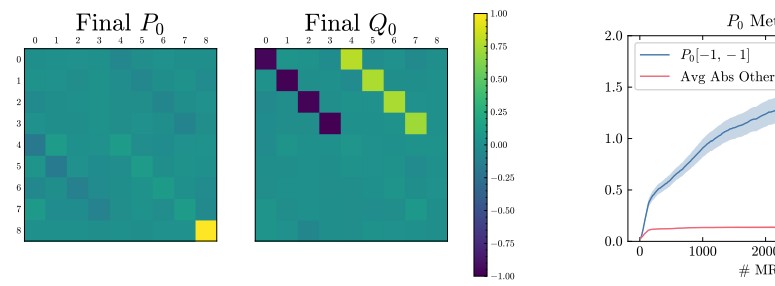

(a) Learned $P_0$ and $Q_0$ after 4000 MRPs

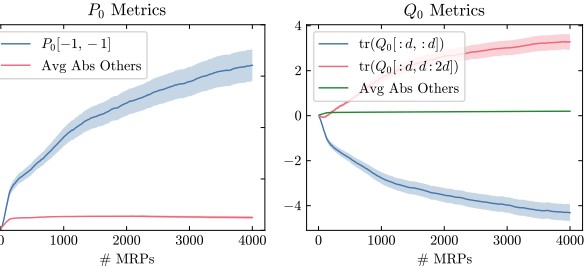

(b) Element-wise learning progress of $P_0$ and $Q_0$

Figure 1: Visualization of the learned transformers and the learning progress. Both (a) and (b) are averaged across 30 seeds and the shaded region in (b) denotes the standard errors. Since $P_0$ and $Q_0$ are in the same product in (3), the algorithm can rescale both or flip the sign of both, but still end up with exactly the same transformer. Therefore, to make sure the visualization are informative, we rescale $P_0$ and $Q_0$ properly first before visualization. See Appendix C.1.1 for details.

The proof is in A.5 with numerical verification in Appendix E as a sanity check.

**Average-Reward TD.** We now demonstrate that transformers are expressive enough to implement in-context average-reward TD. Different from TD(0), average-reward TD exhibits additional challenges in that it updates two estimates (i.e., $w_t$ and $\bar{r}_t$) in parallel. To account for this challenge, we use two additional mechanisms beyond the naive single-head linear transformer. Namely, we allow additional "memory" in the prompt and consider two-head linear transformers. Given a trajectory $(S_0, R_1, S_1, R_2, S_3, R_4, \ldots, S_n)$ sampled from an MRP, we construct the prompt matrix $Z_0$ as

$$Z_0 = \begin{bmatrix} \phi_0 & \cdots & \phi_{n-1} & \phi_n \\ \phi_1 & \cdots & \phi_n & 0 \\ R_1 & \cdots & R_n & 0 \\ 0 & \cdots & 0 & 0 \end{bmatrix} \in \mathbb{R}^{(2d+2)\times(n+1)}.$$

Notably, the last row of zeros is the "memory", which is used by the transformer to store some intermediate quantities during the inference time. We then define the transformer parameters and masks as

$$P_l^{\overline{\mathrm{TD}},(1)} \doteq \begin{bmatrix} 0_{2d\times 2d} & 0_{2d\times 1} & 0_{2d\times 1} \\ 0_{1\times 2d} & 1 & 0 \\ 0_{1\times 2d} & 0 & 0 \end{bmatrix},$$

$$P_l^{\overline{\mathrm{TD}},(2)} \doteq \begin{bmatrix} 0_{2d\times 2d} & 0_{2d\times 1} & 0_{2d\times 1} \\ 0_{1\times 2d} & 0 & 0 \\ 0_{1\times 2d} & 0 & 1 \end{bmatrix}, \quad (18)$$

$$Q_l^{\overline{\mathrm{TD}}} \doteq \begin{bmatrix} -C_l^\top & C_l^\top & 0_{d\times 2} \\ 0_{d\times d} & 0_{d\times d} & 0_{d\times 2} \\ 0_{2\times d} & 0_{2\times d} & 0_{2\times 2} \end{bmatrix},$$

$$W_l \doteq \begin{bmatrix} 0_{2d\times 2d} & 0_{2d\times 1} & 0_{2d\times(2d+2)} & 0_{2d\times 1} \\ 0_{1\times 2d} & 1 & 0_{1\times(2d+2)} & 1 \end{bmatrix}, \quad (19)$$

$$M^{\overline{\mathrm{TD}},(2)} \doteq \begin{bmatrix} I_n & 0_{n\times 1} \\ 0_{1\times n} & 0 \end{bmatrix},$$

$$M^{\overline{\mathrm{TD}},(1)} \doteq \left(I_{n+1} - U_{n+1}\mathrm{diag}\left(\begin{bmatrix} 1 & \frac{1}{2} & \cdots & \frac{1}{n+1} \end{bmatrix}\right)\right) \cdot M^{\overline{\mathrm{TD}},(2)}, \quad (20)$$

where $C_l \in \mathbb{R}^{d\times d}$ is again an arbitrary matrix, $U_{n+1}$ is the $(n+1)\times(n+1)$ upper triangle matrix where all the nonzero elements are 1, and $\mathrm{diag}(x)$ constructs a diagonal matrix with the diagonal entry being $x$. Here, $\left\{P_l^{\overline{\mathrm{TD}},(1)}, Q_l^{\overline{\mathrm{TD}}}\right\}$ are the parameters of the first attention heads, with the input mask being $M^{\overline{\mathrm{TD}},(1)}$. $\left\{P_l^{\overline{\mathrm{TD}},(2)}, Q_l^{\overline{\mathrm{TD}}}\right\}$ are the parameters of the second attention heads, with the input mask being $M^{\overline{\mathrm{TD}},(2)}$. The two heads coincide on some parameters. $W_l$ is the affine transformation that combines the embeddings from the two attention heads. Define the two-head linear-attention as

$$\mathrm{TwoHead}(Z; P, Q, M, P', Q', M', W)$$
$$\doteq W\begin{bmatrix} \mathrm{LinAttn}(Z; P, Q, M) \\ \mathrm{LinAttn}(Z; P', Q', M') \end{bmatrix}.$$

The $L$-layer transformer we are interested in is then given by

$$Z_{l+1} \doteq Z_l + \frac{1}{n}\mathrm{TwoHead}(Z_l; P_l^{\overline{\mathrm{TD}},(1)}, Q_l^{\overline{\mathrm{TD}}}, M^{\overline{\mathrm{TD}},(1)},$$
$$P_l^{\overline{\mathrm{TD}},(2)}, Q_l^{\overline{\mathrm{TD}}}, M^{\overline{\mathrm{TD}},(2)}, W_l). \quad (21)$$

**Theorem 5.3** (Forward pass as average-reward TD)**.** *Consider the $L$-layer transformer in (21). Let $h_l^{(n+1)}$ be the bottom-right element of the $l$-th layer output, i.e., $h_l^{(n+1)} \doteq Z_l[2d+2, n+1]$. Then, it holds that $h_l^{(n+1)} = -\langle\phi_n, w_l\rangle$ where $\{w_l\}$ is defined as $w_0 = 0$,*

$$w_{l+1} = w_l + \frac{1}{n}C_l\sum_{j=1}^{n}\left(R_j - \bar{r}_j + w_l^\top\phi_j - w_l^\top\phi_{j-1}\right)\phi_{j-1}$$

*for $l = 0, \ldots, L-1$, where $\bar{r}_j \doteq \frac{1}{j} \sum_{k=1}^{j} R_k$.*

The proof is in A.6 with numerical verification in Appendix E as a sanity check.

## 6. Related Works

**In-Context Learning.** Understanding in-context learning empirically and theoretically has recently emerged as an active research area (Garg et al., 2022; Müller et al., 2022; Akyürek et al., 2023; von Oswald et al., 2023; Zhao et al., 2023; Allen-Zhu & Li, 2023; Zhang et al., 2023; Mahankali et al., 2023; Ahn et al., 2024), building on prior research demonstrating that neural networks are able to implement algorithms (Siegelmann & Sontag, 1992; Graves et al., 2014; Jastrzębski et al., 2017) and achieve meta-learning from the inputs (Hochreiter et al., 2001). This work advances this line of research by demonstrating **how transformers implement in-context TD**, accompanied by a **theoretical understanding of its emergence**.

**In-Context Reinforcement Learning.** Existing research on in-context RL predominantly adopts a *policy-based* approach, often relying on *supervised pre-training* (Laskin et al., 2022; Raparthy et al., 2023; Sinii et al., 2023; Zisman et al., 2023; Krishnamurthy et al., 2024). Transformers are trained to output the action, instead of the value, for the query state. Correspondingly, the prompts used in this setup consist of previous trajectories from an MDP

$$\underbrace{S_0 A_0 R_1 S_1 A_2 R_2 \ldots S_{t-1} A_{t-1}}_{\text{prompt}} \underbrace{S_t}_{\text{query}} \rightarrow \underbrace{A_t}_{\text{output}}.$$

The dataset usually consists of multiple such prompt-query-output pairs, where *maximum likelihood estimation* is essentially used to train the transformers. Notably, the prompt can be generated by following multiple policies. The prompt can also be offline data containing all trajectories generated during prior RL algorithm training across multiple episodes. This line of research is closely related to offline policy distillation, the goal of which is to learn a policy from offline data using transformers (Chen et al., 2021; Janner et al., 2021; Lee et al., 2022; Reed et al., 2022; Kirsch et al., 2023). Despite that empirical successes observed in the work above, theoretical analysis is often missing. (Lin et al., 2023) provide theoretical analysis for this policy-based supervised pre-training approach and show that the transformers can **approximate** a few RL algorithms, including LinUCB (Chu et al., 2011) and Thompson sampling (Russo et al., 2018) for linear bandits (Lattimore & Szepesvári, 2020) and UCB-VI (Azar et al., 2017) for MDPs. Specifically, (Lin et al., 2023) prove the inference process of the learned transformers **behaves** similarly to those aforementioned RL algorithms in terms of action selection probabilities, regret, and other metrics. This behavioral similarity is also investigated in Lee

et al. (2024). However, the underlying mechanisms within the learned transformers that induce this similarity remains unclear. In contrast, **we go beyond behavioral similarity and prove that transformers can exactly implement a few RL algorithms in its forward pass**. Moreover, we do not use the supervised pre-training paradigm, which is centered on maximum likelihood estimation. As shown in Algorithm 1, we instead use RL pre-training predicated on TD, a *value-based* method. Park et al. (2024) concurrently use a regret-based loss for training transformers in online learning. Brooks et al. (2024) implement policy iteration, a value-based strategy, with transformers, but perform the required $\arg\max$ operation outside the transformers. Despite the observed empirical success, Brooks et al. (2024) also lack a theoretical analysis of their approach.

**Meta-Learning of RL algorithms.** Our Algorithm 1 can be regarded as a meta RL algorithm (Beck et al., 2023), where $d_{\text{task}}$ is the task distribution in the meta RL framework. The learned transformers can be regarded as a learned algorithm, which is used to solve new evaluation tasks from the task distribution. Such meta learning of RL algorithms has been explored in (Duan et al., 2016; Wang et al., 2016; Finn et al., 2017; Kirsch et al., 2019; Oh et al., 2020; Lu et al., 2022; Kirsch et al., 2022; Lu et al., 2023). However, those discovered algorithms lack interpretability – it is not clear *how* the neural network implements the discovered algorithms. By contrast, the discovered transformer from Algorithm 1 is well explained.

## 7. Conclusion

This work demonstrates that transformers **can** and **do** learn to implement temporal difference methods for in-context policy evaluation in the forward pass. We further provide a theoretical explanation of how in-context TD emerges by characterizing an invariant set of the multi-task TD algorithm used in pre-training, bridging the gap between "can" and "do". However, there are a few limitations. First, this work is focused on policy evaluation, with control algorithms deferred to future research. Second, the analysis is largely theoretical – we leave the large-scale verification of the multi-task TD pre-training paradigm for future work. Third, the theoretical analysis of the pre-training paradigm is confined to single-layer linear transformers, leaving the exploration of multi-layer softmax transformers for future studies. In conclusion, this research aims to illuminate the mechanisms of in-context learning, and motivate further investigation into in-context value-based RL.

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

# A. Proofs

## A.1. Proof of Theorem 3.1

*Proof.* We recall from (5) that the embedding evolves according to

$$Z_{l+1} = Z_l + \frac{1}{n} P_l Z_l M(Z_l^\top Q_l Z_l).$$

We first express $Z_l$ using elements of $Z_0$. To this end, it is convenient to give elements of $Z_l$ different names, in particular, we refer to the elements in $Z_l$ as $\left\{ (x_l^{(i)}, y_l^{(i)}) \right\}_{i=1,\dots,n+1}$ in the following way

$$Z_l = \begin{bmatrix} x_l^{(1)} & \cdots & x_l^{(n)} & x_l^{(n+1)} \\ y_l^{(1)} & \cdots & y_l^{(n)} & y_l^{(n+1)} \end{bmatrix},$$

where we recall that $Z_l \in \mathbb{R}^{(2d+1)\times(n+1)}, x_l^{(i)} \in \mathbb{R}^{2d}, y_l^{(i)} \in \mathbb{R}$. Sometimes it is more convenient to refer to the first half and second half of $x_l^{(i)}$ separately, by, e.g., $\nu_l^{(i)} \in \mathbb{R}^d, \xi_l^{(i)} \in \mathbb{R}^d$, i.e., $x_l^{(i)} = \begin{bmatrix} \nu_l^{(i)} \\ \xi_l^{(i)} \end{bmatrix}$. Then we have

$$Z_l = \begin{bmatrix} \nu_l^{(1)} & \cdots & \nu_l^{(n)} & \nu_l^{(n+1)} \\ \xi_l^{(1)} & \cdots & \xi_l^{(n)} & \xi_l^{(n+1)} \\ y_l^{(1)} & \cdots & y_l^{(n)} & y_l^{(n+1)} \end{bmatrix}.$$

We utilize the shorthands

$$X_l = \begin{bmatrix} x_l^{(1)} & \cdots & x_l^{(n)} \end{bmatrix} \in \mathbb{R}^{2d\times n},$$

$$Y_l = \begin{bmatrix} y_l^{(1)} & \cdots & y_l^{(n)} \end{bmatrix} \in \mathbb{R}^{1\times n}.$$

Then we have

$$Z_l = \begin{bmatrix} X_l & x_l^{(n+1)} \\ Y_l & y_l^{(n+1)} \end{bmatrix}.$$

For the input $Z_0$, we assume $\xi_0^{(n+1)} = 0, y_0^{(n+1)} = 0$ but all other entries of $Z_0$ are arbitrary. We recall our definition of $M$ in (4) and $\{P_l^{\mathrm{TD}}, Q_l^{\mathrm{TD}}\}_{l=0,\dots,L-1}$ in (10). In particular, we can express $Q_l^{\mathrm{TD}}$ in a more compact way as

$$M_1 \doteq \begin{bmatrix} -I_d & I_d \\ 0_{d\times d} & 0_{d\times d} \end{bmatrix} \in \mathbb{R}^{2d\times 2d},$$

$$B_l \doteq \begin{bmatrix} C_l^\top & 0_{d\times d} \\ 0_{d\times d} & 0_{d\times d} \end{bmatrix} \in \mathbb{R}^{2d\times 2d},$$

$$A_l \doteq B_l M_1 = \begin{bmatrix} -C_l^\top & C_l^\top \\ 0_{d\times d} & 0_{d\times d} \end{bmatrix} \in \mathbb{R}^{2d\times 2d},$$

$$Q_l^{\mathrm{TD}} \doteq \begin{bmatrix} A_l & 0_{2d\times 1} \\ 0_{1\times 2d} & 0 \end{bmatrix} \in \mathbb{R}^{(2d+1)\times(2d+1)}.$$

We now proceed with the following claims.

**Claim 1.** $X_l \equiv X_0, x_l^{(n+1)} \equiv x_0^{(n+1)}, \forall l.$

Recall that $P_l^{\mathrm{TD}} \doteq \begin{bmatrix} 0_{2d\times 2d} & 0_{2d\times 1} \\ 0_{1\times 2d} & 1 \end{bmatrix} \in \mathbb{R}^{(2d+1)\times(2d+1)}$. Let

$$W_l \doteq Z_l M\left(Z_l^\top Q_l^{\mathrm{TD}} Z_l\right) \in \mathbb{R}^{(2d+1)\times(n+1)}.$$

The embedding evolution can then be expressed as

$$Z_{l+1} = Z_l + \frac{1}{n} P_l^{\text{TD}} W_l.$$

By simple matrix arithmetic, we get

$$P_l^{\text{TD}} W_l = \begin{bmatrix} 0_{2d \times (n+1)} \\ W_l(2d+1) \end{bmatrix},$$

where $W_l(2d+1)$ denotes the $(2d+1)$-th row of $W_l$. Therefore, we have $X_{l+1} = X_l, x_{l+1}^{(n+1)} = x_l^{(n+1)}$. By induction, we get $X_l \equiv X_0$ and $x_l^{(n+1)} \equiv x_0^{(n+1)}$ for all $l = [0, \ldots, L-1]$.

In light of this, we drop all the subscripts of $X_l$, as well as subscripts of $x_l^{(i)}$ for $i = 1, \ldots, n+1$.

**Claim 2.**

$$Y_{l+1} = Y_l + \frac{1}{n} Y_l X^\top A_l X$$

$$y_{l+1}^{(n+1)} = y_l^{(n+1)} + \frac{1}{n} Y_l X^\top A_l x^{(n+1)}.$$

The easier way to show why this claim holds is to factor the embedding evolution into the product of $P_l^{\text{TD}} Z_l M$ and $Z_l^\top Q_l^{\text{TD}} Z_l$. Firstly, we have

$$P_l^{\text{TD}} Z_l = \begin{bmatrix} 0_{2d \times n} & 0_{2d \times 1} \\ Y_l & y_l^{(n+1)} \end{bmatrix}.$$

Applying the mask, we get

$$P_l^{\text{TD}} Z_l M = \begin{bmatrix} 0_{2d \times n} & 0_{2d \times 1} \\ Y_l & 0 \end{bmatrix}.$$

Then, we analyze $Z_l^\top Q_l^{\text{TD}} Z_l$. Applying the block matrix notations, we get

$$\begin{aligned}
Z_l^\top Q_l^{\text{TD}} Z_l &= \begin{bmatrix} X^\top & Y_l^\top \\ x^{(n+1)^\top} & y_l^{(n+1)} \end{bmatrix} \begin{bmatrix} A_l & 0_{2d \times 1} \\ 0_{1 \times 2d} & 0 \end{bmatrix} \begin{bmatrix} X & x^{(n+1)} \\ Y_l & y_l^{(n+1)} \end{bmatrix} \\
&= \begin{bmatrix} X^\top A_l & 0_{n \times 1} \\ x^{(n+1)^\top} A_l & 0 \end{bmatrix} \begin{bmatrix} X & x^{(n+1)} \\ Y_l & y_l^{(n+1)} \end{bmatrix} \\
&= \begin{bmatrix} X^\top A_l X & X^\top A_l x^{(n+1)} \\ x^{(n+1)^\top} A_l X & x^{(n+1)^\top} A_l x^{(n+1)} \end{bmatrix}.
\end{aligned}$$

Combining the two, we get

$$\begin{aligned}
P_l^{\text{TD}} Z_l M \left( Z_l^\top Q_l^{\text{TD}} Z_l \right) &= \begin{bmatrix} 0_{2d \times n} & 0_{2d \times 1} \\ Y_l & 0 \end{bmatrix} \begin{bmatrix} X^\top A_l X & X^\top A_l x^{(n+1)} \\ x^{(n+1)^\top} A_l X & x^{(n+1)^\top} A_l x^{(n+1)} \end{bmatrix} \\
&= \begin{bmatrix} 0_{2d \times n} & 0_{2d \times 1} \\ Y_l X^\top A_l X & Y_l X^\top A_l x^{(n+1)} \end{bmatrix}.
\end{aligned}$$

Hence, according to our update rule in (5), we get

$$Y_{l+1} = Y_l + \frac{1}{n} Y_l X^\top A_l X$$

$$y_{l+1}^{(n+1)} = y_l^{(n+1)} + \frac{1}{n} Y_l X^\top A_l x^{(n+1)}.$$

**Claim 3.**

$$y_{l+1}^{(i)} = y_0^{(i)} + \left\langle M_1 x^{(i)}, \frac{1}{n} \sum_{j=0}^{l} B_j^\top M_2 X Y_j^\top \right\rangle,$$

for $i = 1, \ldots, n+1$, where $M_2 = \begin{bmatrix} I_d & 0_{d \times d} \\ 0_{d \times d} & 0_{d \times d} \end{bmatrix}$.

Following Claim 2, we can unroll $Y_{l+1}$ as

$$Y_{l+1} = Y_l + \frac{1}{n} Y_l X^\top A_l X$$

$$Y_l = Y_{l-1} + \frac{1}{n} Y_{l-1} X^\top A_{l-1} X$$

$$\vdots$$

$$Y_1 = Y_0 + \frac{1}{n} Y_0 X^\top A_0 X.$$

We can then compactly express $Y_{l+1}$ as

$$Y_{l+1} = Y_0 + \frac{1}{n} \sum_{j=0}^{l} Y_j X^\top A_j X.$$

Recall that we define $A_j = B_j M_1$. Then, we can rewrite $Y_{l+1}$ as

$$Y_{l+1} = Y_0 + \frac{1}{n} \sum_{j=0}^{l} Y_j X^\top M_2 B_j M_1 X.$$

The introduction of $M_2$ here does not break the equivalence because $B_j = M_2 B_j$. However, it will help make our proof steps easier to comprehend later.

With the identical procedure, we can easily rewrite $y_{l+1}^{(n+1)}$ as

$$y_{l+1}^{(n+1)} = y_0^{(n+1)} + \frac{1}{n} \sum_{j=0}^{l} Y_j X^\top M_2 B_j M_1 x^{(n+1)}.$$

In light of this, we define $\psi_0 \doteq 0$ and for $l = 0, \ldots$

$$\psi_{l+1} \doteq \frac{1}{n} \sum_{j=0}^{l} B_j^\top M_2 X Y_j^\top \in \mathbb{R}^{2d}. \tag{22}$$

Then we can write

$$y_{l+1}^{(i)} = y_0^{(i)} + \left\langle M_1 x^{(i)}, \psi_{l+1} \right\rangle, \tag{23}$$

for $i = 1, \ldots, n+1$, which is the claim we made. In particular, since we assume $y_0^{(n+1)} = 0$, we have

$$y_{l+1}^{(n+1)} = \left\langle M_1 x^{(n+1)}, \psi_{l+1} \right\rangle.$$

**Claim 4.** The bottom $d$ elements of $\psi_l$ are always 0, i.e., there exists a sequence $\{w_l \in \mathbb{R}^d\}$ such that we can express $\psi_l$ as

$$\psi_l = \begin{bmatrix} w_l \\ 0_{d \times 1} \end{bmatrix}. \tag{24}$$

for all $l = 0, 1, \ldots, L$.

We prove the claim by induction. The base case holds trivially since $\psi_0 \doteq 0$. Suppose that for some $l$, (24) holds. It can be easily verified from the definition of $\psi_{l+1}$ in (22) that

$$\psi_{l+1} = \psi_l + \frac{1}{n} B_l^\top M_2 X Y_l^\top. \tag{25}$$

If we let

$$N_l = \frac{1}{n} M_2 X Y_l^\top \in \mathbb{R}^{2d \times 1},$$

the evolution of $\psi_{l+1}$ can then be compactly expressed as,

$$\psi_{l+1} = \psi_l + B_l^\top N_l.$$

By matrix arithmetic, we have

$$B_l^\top N_l = \begin{bmatrix} C_l^\top & 0_{d \times d} \\ 0_{d \times d} & 0_{d \times d} \end{bmatrix}^\top \begin{bmatrix} N_l(1:d) \\ N_l(d:2d) \end{bmatrix}$$
$$= \begin{bmatrix} C_l N_l(1:d) \\ 0_{d \times 1} \end{bmatrix}$$

where $N_l(1:d) \in \mathbb{R}^d$ and $N_l(d:2d) \in \mathbb{R}^d$ represent the first $d$ and second $d$ elements of $N_l$ respectively. Substituting in our inductive hypothesis into (25), we have:

$$\psi_{l+1} = \begin{bmatrix} w_l \\ 0_{d \times 1} \end{bmatrix} + \begin{bmatrix} C_l N_l(1:d) \\ 0_{d \times 1} \end{bmatrix},$$
$$= \begin{bmatrix} w_l + C_l N_l(1:d) \\ 0_{d \times 1} \end{bmatrix}$$

if we let $w_{l+1} = w_l + C_l N_l(1:d)$, we can see that the property holds for $\psi_{l+1}$, thereby verifying Claim 4.

Given all the claims above, we can then compute that

$$\left\langle \psi_{l+1}, M_1 x^{(n+1)} \right\rangle$$
$$= \left\langle \psi_l, M_1 x^{(n+1)} \right\rangle + \frac{1}{n} \left\langle B_l^\top M_2 X Y_l^\top, M_1 x^{(n+1)} \right\rangle \tag{By (25)}$$
$$= \left\langle \psi_l, M_1 x^{(n+1)} \right\rangle + \frac{1}{n} \sum_{i=1}^n \left\langle B_l^\top M_2 x^{(i)} y_l^{(i)}, M_1 x^{(n+1)} \right\rangle$$
$$= \left\langle \psi_l, M_1 x^{(n+1)} \right\rangle + \frac{1}{n} \sum_{i=1}^n \left\langle B_l^\top M_2 x^{(i)} \left( \left\langle \psi_l, M_1 x^{(i)} \right\rangle + y_0^{(i)} \right), M_1 x^{(n+1)} \right\rangle \tag{By (23)}$$
$$= \left\langle \psi_l, M_1 x^{(n+1)} \right\rangle + \frac{1}{n} \sum_{i=1}^n \left\langle B_l^\top \begin{bmatrix} \nu^{(i)} \\ 0_{d \times 1} \end{bmatrix} \left( \left\langle \psi_l, \begin{bmatrix} -\nu^{(i)} + \xi^{(i)} \\ 0_{d \times 1} \end{bmatrix} \right\rangle + y_0^{(i)} \right), M_1 x^{(n+1)} \right\rangle$$
$$= \left\langle \psi_l, M_1 x^{(n+1)} \right\rangle + \frac{1}{n} \sum_{i=1}^n \left\langle \begin{bmatrix} C_l \nu^{(i)} \\ 0_{d \times 1} \end{bmatrix} \left( y_0^{(i)} + w_l^\top \xi^{(i)} - w_l^\top \nu^{(i)} \right), M_1 x^{(n+1)} \right\rangle \tag{By Claim 4}$$
$$= \left\langle \psi_l, M_1 x^{(n+1)} \right\rangle + \frac{1}{n} \sum_{i=1}^n \left\langle \begin{bmatrix} C_l \nu^{(i)} \left( y_0^{(i)} + w_l^\top \xi^{(i)} - w_l^\top \nu^{(i)} \right) \\ 0_{d \times 1} \end{bmatrix}, M_1 x^{(n+1)} \right\rangle$$

This means

$$\left\langle w_{l+1}, \nu^{(n+1)} \right\rangle = \left\langle w_l, \nu^{(n+1)} \right\rangle + \frac{1}{n} \sum_{i=1}^n \left\langle C_l \nu^{(i)} \left( y_0^{(i)} + w_l^\top \xi^{(i)} - w_l^\top \nu^{(i)} \right), \nu^{(n+1)} \right\rangle.$$

Since the choice of the query $\nu^{(n+1)}$ is arbitrary, we get

$$w_{l+1} = w_l + \frac{1}{n}\sum_{i=1}^{n} C_l\Big(y_0^{(i)} + w_l^\top \xi^{(i)} - w_l^\top \nu^{(i)}\Big)\nu^{(i)}.$$

In particular, when we construct $Z_0$ such that $\nu^{(i)} = \phi_{i-1}$, $\xi^{(i)} = \gamma\phi_i$ and $y_0^{(i)} = R_i$, we get

$$w_{l+1} = w_l + \frac{1}{n}\sum_{i=1}^{n} C_l\big(R_i + \gamma w_l^\top \phi_i - w_l^\top \phi_{i-1}\big)\phi_{i-1}$$

which is the update rule for pre-conditioned TD learning. We also have

$$y_l^{(n+1)} = \Big\langle \psi_l, M_1 x^{(n+1)} \Big\rangle = -\Big\langle w_l, \phi^{(n+1)} \Big\rangle.$$

This concludes our proof. $\qquad\square$

### A.2. Proof of Corollary 3.2

*Proof.* The proof presented here closely mirrors the methodology and notation established in Theorem 3.1. Since we are only considering a 1-layer transformer in this Corollary, we can recall the embedding evolution from (5) and write

$$Z_1 = Z_0 + \frac{1}{n}P_0 Z_0 M(Z_0^\top Q_0 Z_0).$$

We once again refer to the elements in $Z_l$ as $\Big\{(x_l^{(i)}, y_l^{(i)})\Big\}_{i=1,\dots,n+1}$ in the following way

$$Z_l = \begin{bmatrix} x_l^{(1)} & \cdots & x_l^{(n)} & x_l^{(n+1)} \\ y_l^{(1)} & \cdots & y_l^{(n)} & y_l^{(n+1)} \end{bmatrix},$$

where we recall that $Z_l \in \mathbb{R}^{(2d+1)\times(n+1)}, x_l^{(i)} \in \mathbb{R}^{2d}, y_l^{(i)} \in \mathbb{R}$. We utilize, $\nu_l^{(i)} \in \mathbb{R}^d, \xi_l^{(i)} \in \mathbb{R}^d$, to refer to the first half and second half of $x_l^{(i)}$ i.e., $x_l^{(i)} = \begin{bmatrix} \nu_l^{(i)} \\ \xi_l^{(i)} \end{bmatrix}$. Then we have

$$Z_l = \begin{bmatrix} \nu_l^{(1)} & \cdots & \nu_l^{(n)} & \nu_l^{(n+1)} \\ \xi_l^{(1)} & \cdots & \xi_l^{(n)} & \xi_l^{(n+1)} \\ y_l^{(1)} & \cdots & y_l^{(n)} & y_l^{(n+1)} \end{bmatrix}.$$

We further define as shorthands

$$X_l = \begin{bmatrix} x_l^{(1)} & \cdots & x_l^{(n)} \end{bmatrix} \in \mathbb{R}^{2d\times n}, \ Y_l = \begin{bmatrix} y_l^{(1)} & \cdots & y_l^{(n)} \end{bmatrix} \in \mathbb{R}^{1\times n}.$$

Then the blockwise structure of $Z_l$ can be succinctly expressed as:

$$Z_l = \begin{bmatrix} X_l & x_l^{(n+1)} \\ Y_l & y_l^{(n+1)} \end{bmatrix}.$$

For the input $Z_0$, we assume $\xi_0^{(n+1)} = 0, y_0^{(n+1)} = 0$ but all other entries of $Z_0$ are arbitrary. We recall our definition of $M$ in (4) and $\{P_0, Q_0\}$ in (10). In particular, we can express $Q_0$ in a more compact way as

$$M_1 \doteq \begin{bmatrix} -I_d & 0_{d\times d} \\ 0_{d\times d} & 0_{d\times d} \end{bmatrix} \in \mathbb{R}^{2d\times 2d}, \ B_0 \doteq \begin{bmatrix} C_0^\top & 0_{d\times d} \\ 0_{d\times d} & 0_{d\times d} \end{bmatrix} \in \mathbb{R}^{2d\times 2d},$$

$$A_0 \doteq B_0 M_1 = \begin{bmatrix} -C_0^\top & 0_{d\times d} \\ 0_{d\times d} & 0_{d\times d} \end{bmatrix} \in \mathbb{R}^{2d\times 2d},$$

$$Q_0 \doteq \begin{bmatrix} A_0 & 0_{2d\times 1} \\ 0_{1\times 2d} & 0 \end{bmatrix} \in \mathbb{R}^{(2d+1)\times(2d+1)}.$$

We will proceed with the following claims.

**Claim 1.** $X_1 \equiv X_0, x_1^{(n+1)} \equiv x_0^{(n+1)}$

Because we are considering the special case of $L = 1$ and because we utilize the same definition of $P_0$ as in Theorem 3.1, the argument proving Claim 1 in Theorem 3.1 holds here as well. As a result, we drop all the subscripts of $X_1$, as well as subscripts of $x_1^{(i)}$ for $i = 1, \ldots, n + 1$.

**Claim 2.**

$$Y_1 = Y_0 + \frac{1}{n} Y_0 X^\top A_0 X$$
$$y_1^{(n+1)} = y_0^{(n+1)} + \frac{1}{n} Y_0 X^\top A_0 x^{(n+1)}.$$

This claim is a special case of Claim 2 from the proof of Theorem 3.1 in Appendix A.1, where $L = 1$. Our block-wise construction of $Q_0$ matches that in the proof of Theorem 3.1. Although our $A_0$ here differs from the specific form of $A_0$ in the proof of Theorem 3.1, this specific form is not utilized in the proof of Claim 2. Therefore, the proof of Claim 2 in Appendix A.1 applies here, and we omit the steps to avoid redundancy.

**Claim 3.**

$$y_1^{(i)} = y_0^{(i)} + \left\langle M_1 x^{(i)}, \frac{1}{n} B_0^\top M_2 X Y_0^\top \right\rangle,$$

for $i = 1, \ldots, n + 1$, where $M_2 = \begin{bmatrix} I_d & 0_{d \times d} \\ 0_{d \times d} & 0_{d \times d} \end{bmatrix}$.

This claim once again is the $L = 1$ case of Claim 3 from the proof of Theorem 3.1 in Appendix A.1. The specific form of $M_1$ is not utilized in the proof of Claim 3 from Appendix A.1, so it applies here.

We can then define $\psi_0 \doteq 0$ and,

$$\psi_1 \doteq \frac{1}{n} B_0^\top M_2 X Y_0^\top \in \mathbb{R}^{2d}. \tag{26}$$

Then we can write

$$y_1^{(i)} = y_0^{(i)} + \left\langle M_1 x^{(i)}, \psi_1 \right\rangle,$$

for $i = 1, \ldots, n + 1$, which is the claim we made. In particular, since we assume $y_0^{(n+1)} = 0$, we have

$$y_1^{(n+1)} = \left\langle M_1 x^{(n+1)}, \psi_1 \right\rangle.$$

**Claim 4.** The bottom $d$ elements of $\psi_1$ are always 0, i.e., there exists $w_1 \in \mathbb{R}^d$ such that we can express $\psi_1$ as

$$\psi_1 = \begin{bmatrix} w_1 \\ 0_{d \times 1} \end{bmatrix}.$$

Since our $B_0$ here is identical to that in the proof of Theorem 3.1 in A.1, Claim 4 holds for the same reason. We therefore omit the proof details to avoid repetition.

Given all the claims above, we can then compute that

$$
\begin{aligned}
\left\langle \psi_1, M_1 x^{(n+1)} \right\rangle &= \frac{1}{n} \left\langle B_0^\top M_2 X Y_0^\top, M_1 x^{(n+1)} \right\rangle && \text{(By (26))} \\
&= \frac{1}{n} \sum_{i=1}^n \left\langle B_0^\top M_2 x^{(i)} y_0^{(i)}, M_1 x^{(n+1)} \right\rangle \\
&= \frac{1}{n} \sum_{i=1}^n \left\langle B_0^\top \begin{bmatrix} \nu^{(i)} \\ 0_{d\times 1} \end{bmatrix} \left( y_0^{(i)} \right), M_1 x^{(n+1)} \right\rangle \\
&= \frac{1}{n} \sum_{i=1}^n \left\langle \begin{bmatrix} C_0 \nu^{(i)} \\ 0_{d\times 1} \end{bmatrix} \left( y_0^{(i)} \right), M_1 x^{(n+1)} \right\rangle && \text{(By Claim 4)} \\
&= \frac{1}{n} \sum_{i=1}^n \left\langle \begin{bmatrix} C_0 \nu^{(i)} y_0^{(i)} \\ 0_{d\times 1} \end{bmatrix}, M_1 x^{(n+1)} \right\rangle
\end{aligned}
$$

This means

$$
\left\langle w_1, \nu^{(n+1)} \right\rangle = \frac{1}{n} \sum_{i=1}^n \left\langle C_0 \nu^{(i)} y_0^{(i)}, \nu^{(n+1)} \right\rangle.
$$

Since the choice of the query $\nu^{(n+1)}$ is arbitrary, we get

$$
w_1 = \frac{1}{n} \sum_{i=1}^n C_0 y_0^{(i)} \nu^{(i)}.
$$

In particular, when we construct $Z_0$ such that $\nu^{(i)} = \phi_{i-1}$ and $y_0^{(i)} = R_i$, we get

$$
w_1 = \frac{1}{n} \sum_{i=1}^n C_0 R_i \phi_{i-1}
$$

which is the update rule for a single step of TD(0) with $w_0 = 0$. We also have

$$
y_1^{(n+1)} = \left\langle \psi_1, M_1 x^{(n+1)} \right\rangle = -\left\langle w_1, \phi^{(n+1)} \right\rangle.
$$

This concludes our proof. □

### A.3. Proof of Theorem 4.3

**Preliminaries** Before we present the proof, we first introduce notations convenient for our analysis. We decompose $P_0$ and $Q_0$ as

$$
P_0 = \begin{bmatrix} P \in \mathbb{R}^{2d\times(2d+1)} \\ p \in \mathbb{R}^{1\times(2d+1)} \end{bmatrix}, Q_0 = \begin{bmatrix} Q_a \in \mathbb{R}^{d\times d} & Q_b \in \mathbb{R}^{d\times d} & q_c \in \mathbb{R}^{d\times 1} \\ Q_a' \in \mathbb{R}^{d\times d} & Q_b' \in \mathbb{R}^{d\times d} & q_c' \in \mathbb{R}^{d\times 1} \\ q_a \in \mathbb{R}^{1\times d} & q_b \in \mathbb{R}^{1\times d} & q_c'' \in \mathbb{R} \end{bmatrix}.
$$

One can readily check that $\text{TF}_1$ is independent of $P, Q_b, Q_b', q_b, q_c, q_c', q_c''$. Thus, we can assume that these matrices are zero. Let $z^{(i)}$ be the $i$-th column of $Z_0$. Indeed, $\text{TF}_1$ can be written as

$$\text{TF}_1(Z_0, \{P_0, Q_0\}) = -Z_1[2d+1, n+1] \qquad \text{(By (6))}$$

$$= -\frac{1}{n} p^\top \left( \sum_{i=1}^n z^{(i)} z^{(i)\top} \right) Q_0 z^{(n+1)}$$

$$= -\frac{1}{n} \sum_{i=1}^n \left\langle p, z^{(i)} \right\rangle z^{(i)\top} Q_0 z^{(n+1)}$$

$$= -\frac{1}{n} \sum_{i=1}^n \left\langle p, z^{(i)} \right\rangle \left( \phi_{i-1}^\top Q_a \phi_{n+1} + \gamma \phi_i^\top Q_a' \phi_{n+1} + R_i \phi_{n+1}^\top q_a \right) \qquad (27)$$

$$= -\frac{1}{n} \sum_{i=1}^n \left( \underbrace{\left\langle p_{[1:d]}, \phi_{i-1} \right\rangle + \gamma \left\langle p_{[d+1:2d]}, \phi_i \right\rangle + p_{[2d+1]} R_i}_{\alpha_i(Z_0, P_0)} \right)$$

$$\cdot \left( \underbrace{\phi_{i-1}^\top Q_a \phi_{n+1} + \gamma (\phi_i)^\top Q_a' \phi_{n+1} + R_i \phi_{n+1}^\top q_a}_{\beta_i(Z_0, Q_0)} \right).$$

We prepare the following gradient computations for future use:

$$\nabla_{p_{[1:d]}} \text{TF}_1(Z_0, \{P_0, Q_0\}) = -\frac{1}{n} \sum_{i=1}^n \beta_i(Z_0, Q_0) \phi_{i-1}$$

$$\nabla_{p_{[d+1:2d]}} \text{TF}_1(Z_0, \{P_0, Q_0\}) = -\frac{\gamma}{n} \sum_{i=1}^n \beta_i(Z_0, Q_0) \phi_i$$

$$\nabla_{Q_a} \text{TF}_1(Z_0, \{P_0, Q_0\}) = -\frac{1}{n} \sum_{i=1}^n \alpha_i(Z_0, P_0) \phi_{i-1} \phi_{n+1}^\top \qquad (28)$$

$$\nabla_{Q_a'} \text{TF}_1(Z_0, \{P_0, Q_0\}) = -\frac{\gamma}{n} \sum_{i=1}^n \alpha_i(Z_0, P_0) \phi_i \phi_{n+1}^\top$$

$$\nabla_{q_a} \text{TF}_1(Z_0, \{P_0, Q_0\}) = -\frac{1}{n} \sum_{i=1}^n R_i \alpha_i(Z_0, P_0) \phi_{n+1}.$$

We will also reference the following two lemmas in our main proof.

**Lemma A.1.** *Let $\Lambda$ be a diagonal matrix whose diagonal elements are i.i.d Rademacher random variables* [2] $\zeta_1, \ldots \zeta_d$. *For any matrix $K \in \mathbb{R}^{d \times d}$, we have that $\mathbb{E}_\Lambda[\Lambda K \Lambda] = \text{diag}(K)$.*

*Proof.* First, we can write $\Lambda K \Lambda$ explicitly as

$$\Lambda K \Lambda = \begin{bmatrix} \zeta_1 & 0 & \cdots & 0 \\ 0 & \zeta_2 & \cdots & 0 \\ \vdots & \vdots & \ddots & \vdots \\ 0 & 0 & \cdots & \zeta_d \end{bmatrix} \begin{bmatrix} k_{11} & k_{12} & \cdots & k_{1d} \\ k_{21} & k_{22} & \cdots & k_{2d} \\ \vdots & \vdots & \ddots & \vdots \\ k_{d1} & k_{d2} & \cdots & k_{dd} \end{bmatrix} \begin{bmatrix} \zeta_1 & 0 & \cdots & 0 \\ 0 & \zeta_2 & \cdots & 0 \\ \vdots & \vdots & \ddots & \vdots \\ 0 & 0 & \cdots & \zeta_d \end{bmatrix}.$$

Using $(\Lambda K \Lambda)_{ij}$ to denote the element in the $i$-th row at column $j$ of $\Lambda K \Lambda$, from elementary matrix multiplication we have

$$(\Lambda K \Lambda)_{ij} = \zeta_i k_{ij} \zeta_j.$$

---

[2] A Rademacher random variable takes values 1 or $-1$, each with an equal probability of 0.5.

When $i \neq j$, $\mathbb{E}[\zeta_i \zeta_j] = \mathbb{E}[\zeta_i]\mathbb{E}[\zeta_j] = 0$ becasue $\zeta_i$ and $\zeta_j$ are independent. For $i = j$, $\mathbb{E}[\zeta_i \zeta_j] = \mathbb{E}[\zeta_i^2] = 1$. We can then compute the expectation

$$\mathbb{E}_\Lambda[(\Lambda K \Lambda)]_{ij} = \begin{cases} k_{ij} & i = j \\ 0 & i \neq j. \end{cases}$$

Consequently,

$$\mathbb{E}_\Lambda[\Lambda K \Lambda] = \text{diag}(K).$$

$\square$

**Lemma A.2.** *Let $\Pi \in \mathbb{R}^{d \times d}$ be a random permutation matrix uniformly distributed over all $d \times d$ permutation matrices and $L \in \mathbb{R}^{d \times d}$ be a diagonal matrix. Then, it holds that*

$$\mathbb{E}_\Pi[\Pi L \Pi^\top] = \frac{1}{d}\text{tr}(L)I_d.$$

*Proof.* By definition,

$$[\Pi L \Pi^\top]_{ij} = \sum_{k=1}^d \Pi_{ik} L_{kk} \Pi_{jk}.$$

We note that each row of $\Pi$ is a standard basis. Given the orthogonality of standard bases, we get

$$[\Pi L \Pi^\top]_{ij} = \begin{cases} 0 & i \neq j \\ L_{q_i q_i} & i = j \end{cases},$$

where $q_i$ is the unique index such that $\Pi_{i q_i} = 1$. If the distribution of $\Pi$ is uniform, then $[\Pi L \Pi^\top]_{ii}$ is equal to one of $L_{11}, \ldots, L_{dd}$ with the same probability. Thus, the expected value $[\Pi L \Pi^\top]_{ii}$ is $\frac{1}{d}\text{tr}(L)$. $\square$

Now, we start with the proof of the theorem statement.

*Proof.* We recall the definition of the set $\Theta^*$ as

$$\Theta^* \doteq \cup_{\eta, c, c' \in \mathbb{R}} \left\{ P = \begin{bmatrix} 0_{2d \times 2d} & 0_{2d \times 1} \\ 0_{1 \times 2d} & \eta \end{bmatrix}, Q = \begin{bmatrix} cI_d & 0_{d \times d} & 0_{d \times 1} \\ c'I_d & 0_{d \times d} & 0_{d \times 1} \\ 0_{1 \times d} & 0_{1 \times d} & 0 \end{bmatrix} \right\}.$$

Suppose $\theta_k \in \Theta^*$, then by (27) and (28), we get

$$\text{TF}_1(Z_0, \theta_k) = -\frac{\eta_k}{n} \sum_{i=1}^n R_i \left( c_k \phi_{i-1}^\top \phi_{n+1} + c_k' \gamma \phi_i^\top \phi_{n+1} \right) \tag{29}$$

$$\text{TF}_1(Z_0', \theta_k) = -\frac{\eta_k}{n} \sum_{i=1}^n R_{i+1} \left( c_k \phi_i^\top \phi_{n+2} + c_k' \gamma \phi_{i+1}^\top \phi_{n+2} \right)$$

$$\nabla_{p_{[1:d]}} \text{TF}_1(Z_0, \theta_k) = -\frac{1}{n} \sum_{i=1}^n \left( c_k \phi_{i-1}^\top \phi_{n+1} + c_k' \gamma \phi_i^\top \phi_{n+1} \right) \phi_{i-1}$$

$$\nabla_{p_{[d+1:2d]}} \text{TF}_1(Z_0, \theta_k) = -\frac{\gamma}{n} \sum_{i=1}^n \left( c_k \phi_{i-1}^\top \phi_{n+1} + c_k' \gamma \phi_i^\top \phi_{n+1} \right) \phi_i$$

$$\nabla_{Q_a} \text{TF}_1(Z_0, \theta_k) = -\frac{\eta_k}{n} \sum_{i=1}^n R_i \phi_{i-1} \phi_{n+1}^\top$$

$$\nabla_{Q_a'} \text{TF}_1(Z_0, \theta_k) = -\frac{\gamma \eta_k}{n} \sum_{i=1}^n R_i \phi_i \phi_{n+1}^\top$$

$$\nabla_{q_a} \text{TF}_1(Z_0, \theta_k) = -\frac{\eta_k}{n} \sum_{i=1}^n R_i^2 \phi_{n+1}$$

Recall the definition of $\Delta(\theta)$ in (13). With a slight abuse of notation, we define $\Delta(p_{[1:d]})$ to be the $p_{[1:d]}$ component of $\Delta(\theta)$, i.e.,

$$\Delta(p_{[1:d]}) \doteq \mathbb{E}\left[(R + \gamma \mathrm{TF}_1(Z'_0, \theta) - \mathrm{TF}_1(Z_0, \theta))\frac{\partial \mathrm{TF}_1(Z_0, \theta)}{\partial p_{[1:d]}}\right].$$

Same goes for $\Delta(p_{[d+1:2d]}), \Delta(Q_a), \Delta(Q'_a)$, and $\Delta(q_a)$.

We will prove that

(a) $\Delta(p_{[1:d]}) = \Delta(p_{[d+1:2d]}) = \Delta(q_a) = 0$ for $\Delta(\theta_k)$;

(b) $\Delta(Q_a) = \delta I_d$ and $\Delta(Q'_a) = \delta' I_d$ for some $\delta, \delta' \in \mathbb{R}$ for $\Delta(\theta_k)$

using Assumptions 4.1 and 4.2. We can see that the combination of (a) and (b) are sufficient for proving the theorem. Recall that $Z_0$ and $Z'_0$ are sampled from $(p_0, p, r, \phi)$. We make the following claims to assist our proof of (a) and (b).

**Claim 1.** Let $\zeta$ be a Rademacher random variable. We denote $Z_\zeta$ and $Z'_\zeta$ as the prompts sampled from $(p_0, p, r, \zeta\phi)$. We then have $Z_0 \triangleq Z_\zeta$ and $Z'_0 \triangleq Z'_\zeta$. To show this is true, we notice that for any realization of $\zeta$, denoted as $\bar{\zeta} \in \{1, -1\}$, we have

$$\begin{aligned}
\Pr(p_0, p, r, \phi) &= \Pr(p_0, p, r)\Pr(\phi) && \text{(Assumption 4.1)}\\
&= \Pr(p_0, p, r)\Pr(\bar{\zeta}I_d\phi) && \text{(Assumption 4.2)}\\
&= \Pr(p_0, p, r, \bar{\zeta}\phi). && \text{(Assumption 4.1)}
\end{aligned}$$

It then follows that

$$\begin{aligned}
\Pr(p_0, p, r, \phi) &= \Pr(p_0, p, r, \phi)\sum_{\bar{\zeta}\in\{1,-1\}}\Pr(\zeta = \bar{\zeta})\\
&= \sum_{\bar{\zeta}\in\{1,-1\}}\Pr(p_0, p, r, \phi)\Pr(\zeta = \bar{\zeta})\\
&= \sum_{\bar{\zeta}\in\{1,-1\}}\Pr(p_0, p, r, \bar{\zeta}\phi)\Pr(\zeta = \bar{\zeta})\\
&= \Pr(p_0, p, r, \zeta\phi).
\end{aligned}$$

This implies Claim 1 holds.

**Claim 2.** Define $\Lambda$ as the diagonal matrix whose diagonal elements are i.i.d. Rademacher random variables $\zeta_1, \ldots, \zeta_d$. We denote $Z_\Lambda$ and $Z'_\Lambda$ as the prompts sampled from $(p_0, p, r, \Lambda\phi)$, where $\Lambda\phi$ means $[\Lambda\phi(s)]_{s\in\mathcal{S}}$. We then have $Z_0 \triangleq Z_\Lambda$ and $Z'_0 \triangleq Z'_\Lambda$. The proof follows the same procedures as Claim 1.

**Claim 3.** Let $\Pi$ be a random permutation matrix uniformly distributed over all $d \times d$ permutation matrices. We denote $Z_\Pi$ and $Z'_\Pi$ as the prompts sampled from $(p_0, p, r, \Pi\phi)$, where $\Pi\phi$ means $[\Pi\phi(s)]_{s\in\mathcal{S}}$. We then have $Z_0 \triangleq Z_\Pi$ and $Z'_0 \triangleq Z'_\Pi$. The proof follows the same procedures as Claim 1.

**Proof of (a) using Claim 1** It is easy to check by (29) that

$$\begin{aligned}
\mathrm{TF}_1(Z_\zeta, \theta_k) &= -\frac{\eta_k}{n}\sum_{i=1}^{n}R_i\left(c_k\zeta^2\phi_{i-1}^\top\phi_{n+1} + c'_k\gamma\zeta^2\phi_i^\top\phi_{n+1}\right)\\
&= \underbrace{\zeta^2}_{=1}\mathrm{TF}_1(Z_0, \theta_k)\\
&= \mathrm{TF}_1(Z_0, \theta_k). && (30)
\end{aligned}$$

Similarly, one can check that $\mathrm{TF}_1(Z'_\zeta, \theta_k) = \mathrm{TF}_1(Z'_0, \theta_k)$.

Furthermore,

$$
\begin{aligned}
\nabla_{p_{[1:d]}} \mathrm{TF}_1(Z_\zeta, \theta_k) &= -\frac{1}{n} \sum_{i=1}^{n} \left( c_k \underbrace{\zeta^2}_{=1} \phi_{i-1}^\top \phi_{n+1} + c_k' \gamma \underbrace{\zeta^2}_{=1} \phi_i^\top \phi_{n+1} \right) \zeta \phi_{i-1} \\
&= -\frac{\zeta}{n} \sum_{i=1}^{n} \left( c_k \phi_{i-1}^\top \phi_{n+1} + c_k' \gamma \phi_i^\top \phi_{n+1} \right) \phi_{i-1} \\
&= \zeta \nabla_{p_{[1:d]}} \mathrm{TF}_1(Z_0, \theta_k).
\end{aligned} \tag{31}
$$

Then, from (13), we get

$$
\begin{aligned}
&\Delta(p_{[1:d]}) \\
&= \mathbb{E}\big[ (R_{n+2} + \gamma \mathrm{TF}_1(Z_0', \theta_k) - \mathrm{TF}_1(Z_0, \theta_k)) \nabla_{p_{[1:d]}} \mathrm{TF}_1(Z_0, \theta_k) \big] \\
&= \mathbb{E}\big[ (R_{n+2} + \gamma \mathrm{TF}_1(Z_\zeta', \theta_k) - \mathrm{TF}_1(Z_\zeta, \theta_k)) \nabla_{p_{[1:d]}} \mathrm{TF}_1(Z_\zeta, \theta_k) \big] && \text{(By Claim 1)} \\
&= \mathbb{E}_\zeta \big[ \mathbb{E}\big[ (R_{n+2} + \gamma \mathrm{TF}_1(Z_\zeta', \theta_k) - \mathrm{TF}_1(Z_\zeta, \theta_k)) \nabla_{p_{[1:d]}} \mathrm{TF}_1(Z_\zeta, \theta_k) \mid \zeta \big] \big] \\
&= \mathbb{E}_\zeta \big[ \mathbb{E}\big[ (R_{n+2} + \gamma \mathrm{TF}_1(Z_0', \theta_k) - \mathrm{TF}_1(Z_0, \theta_k)) \zeta \nabla_{p_{[1:d]}} \mathrm{TF}_1(Z_0, \theta_k) \mid \zeta \big] \big] && \text{(By (30), (31))} \\
&= \mathbb{E}_\zeta \big[ \zeta \mathbb{E}\big[ (R_{n+2} + \gamma \mathrm{TF}_1(Z_0', \theta_k) - \mathrm{TF}_1(Z_0, \theta_k)) \nabla_{p_{[1:d]}} \mathrm{TF}_1(Z_0, \theta_k) \mid \zeta \big] \big] \\
&= \mathbb{E}_\zeta \big[ \zeta \mathbb{E}\big[ (R_{n+2} + \gamma \mathrm{TF}_1(Z_0', \theta_k) - \mathrm{TF}_1(Z_0, \theta_k)) \nabla_{p_{[1:d]}} \mathrm{TF}_1(Z_0, \theta_k) \big] \big] \\
&= \mathbb{E}_\zeta[\zeta] \mathbb{E}\big[ (R_{n+2} + \gamma \mathrm{TF}_1(Z_0', \theta_k) - \mathrm{TF}_1(Z_0, \theta_k)) \nabla_{p_{[1:d]}} \mathrm{TF}_1(Z_0, \theta_k) \big] \\
&= 0.
\end{aligned}
$$

The proof is analogous for $\Delta(p_{[d+1:2d]}) = 0$, and $\Delta(q_a) = 0$.

**Proof of (b) using Claims 2 and 3**  We first show that $\Delta(Q_a)$ is a diagonal matrix. Similar to (a), we have

$$
\begin{aligned}
\mathrm{TF}_1(Z_\Lambda, \theta_k) &= -\frac{1}{n} \sum_{i=1}^{n} \eta_k R_i \left( c_k \phi_{i-1}^\top \underbrace{\Lambda^2}_{=I} \phi_{n+1} + c_k' \gamma \phi_i^\top \underbrace{\Lambda^2}_{=I} \phi_{n+1} \right) \\
&= \mathrm{TF}_1(Z_0, \theta_k).
\end{aligned} \tag{32}
$$

Similarly, we get $\mathrm{TF}_1(Z_\Lambda', \theta_k) = \mathrm{TF}_1(Z_0', \theta_k)$. Additionally, we have

$$
\nabla_{Q_a} \mathrm{TF}_1(Z_\Lambda, \theta_k) = -\frac{1}{n} \sum_{i=1}^{n} \eta_k R_i \Lambda \phi_{i-1} \phi_{n+1}^\top \Lambda^\top = \Lambda \nabla_{Q_a} \mathrm{TF}_1(Z_0, \theta_k) \Lambda. \tag{33}
$$

By (13) again, we get

$$
\begin{aligned}
&\Delta(Q_a) \\
&= \mathbb{E}[(R_{n+2} + \gamma \mathrm{TF}_1(Z_0', \theta_k) - \mathrm{TF}_1(Z_0, \theta_k)) \nabla_{Q_a} \mathrm{TF}_1(Z_0, \theta_k)] \\
&= \mathbb{E}[(R_{n+2} + \gamma \mathrm{TF}_1(Z_\Lambda', \theta_k) - \mathrm{TF}_1(Z_\Lambda, \theta_k)) \nabla_{Q_a} \mathrm{TF}_1(Z_\Lambda, \theta_k)] && \text{(By Claim 2)} \\
&= \mathbb{E}_\Lambda[\mathbb{E}[(R_{n+2} + \gamma \mathrm{TF}_1(Z_\Lambda', \theta_k) - \mathrm{TF}_1(Z_\Lambda, \theta_k)) \nabla_{Q_a} \mathrm{TF}_1(Z_\Lambda, \theta_k) \mid \Lambda]] \\
&= \mathbb{E}_\Lambda[\mathbb{E}[(R_{n+2} + \gamma \mathrm{TF}_1(Z_0', \theta_k) - \mathrm{TF}_1(Z_0, \theta_k)) \Lambda \nabla_{Q_a} \mathrm{TF}_1(Z_0, \theta_k) \Lambda \mid \Lambda]] && \text{(By (32), (33))} \\
&= \mathbb{E}_\Lambda[\Lambda \mathbb{E}[(R_{n+2} + \gamma \mathrm{TF}_1(Z_0', \theta_k) - \mathrm{TF}_1(Z_0, \theta_k)) \nabla_{Q_a} \mathrm{TF}_1(Z_0, \theta_k) \mid \Lambda] \Lambda] \\
&= \mathbb{E}_\Lambda[\Lambda \mathbb{E}[(R_{n+2} + \gamma \mathrm{TF}_1(Z_0', \theta_k) - \mathrm{TF}_1(Z_0, \theta_k)) \nabla_{Q_a} \mathrm{TF}_1(Z_0, \theta_k)] \Lambda] \\
&= \mathrm{diag}(\mathbb{E}[(R_{n+2} + \gamma \mathrm{TF}_1(Z_0', \theta_k) - \mathrm{TF}_1(Z_0, \theta_k)) \nabla_{Q_a} \mathrm{TF}_1(Z_0, \theta_k)]) && \text{(By Lemma A.1)} \\
&= \mathrm{diag}(\Delta(Q_a)).
\end{aligned}
$$

The last equation holds if and only if $\Delta(Q_a)$ is diagonal. We have proven this claim.

Now, we prove that $\Delta(Q_a) = \delta I_d$ for some $\delta \in \mathbb{R}$ using Claim 3 and Lemma A.2. Let $\Pi$ be a random permutation matrix uniformly distributed over all permutation matrices. Recall the definition of $Z_\Pi$ and $Z'_\Pi$ in Claim 3. We have

$$\mathrm{TF}_1(Z_\Pi, \theta_k) = -\frac{1}{n}\sum_{i=1}^{n}\eta_k R_i\left(c_k\phi_{i-1}^\top\underbrace{\Pi^\top\Pi}_{=I}\phi_{n+1} + c'_k\gamma\phi_i^\top\underbrace{\Pi^\top\Pi}_{=I}\phi_{n+1}\right) = \mathrm{TF}_1(Z_0, \theta_k). \tag{34}$$

Analogously, we get $\mathrm{TF}_1(Z'_\Pi, \theta_k) = \mathrm{TF}_1(Z'_0, \theta_k)$. Furthermore, we have

$$\nabla_{Q_a}\mathrm{TF}_1(Z_\Pi, \theta_k) = -\frac{1}{n}\sum_{i=1}^{n}\eta_k R_i\Pi\phi_{i-1}\phi_{n+1}^\top\Pi^\top = \Pi\nabla_{Q_a}\mathrm{TF}_1(Z_0, \theta_k)\Pi^\top. \tag{35}$$

By (13), we are ready to show that

$$
\begin{aligned}
&\Delta(Q_a)\\
=&\mathbb{E}[(R_{n+2} + \gamma\mathrm{TF}_1(Z'_0, \theta_k) - \mathrm{TF}_1(Z_0, \theta_k))\nabla_{Q_a}\mathrm{TF}_1(Z_0, \theta_k)]\\
=&\mathbb{E}[(R_{n+2} + \gamma\mathrm{TF}_1(Z'_\Pi, \theta_k) - \mathrm{TF}_1(Z_\Pi, \theta_k))\nabla_{Q_a}\mathrm{TF}_1(Z_\Pi, \theta_k)] &\text{(By Claim 3)}\\
=&\mathbb{E}_\Pi[\mathbb{E}[(R_{n+2} + \gamma\mathrm{TF}_1(Z'_\Pi, \theta_k) - \mathrm{TF}_1(Z_\Pi, \theta_k))\nabla_{Q_a}\mathrm{TF}_1(Z_\Pi, \theta_k) \mid \Pi]]\\
=&\mathbb{E}_\Pi\big[\mathbb{E}\big[(R_{n+2} + \gamma\mathrm{TF}_1(Z'_0, \theta_k) - \mathrm{TF}_1(Z_0, \theta_k))\Pi\nabla_{Q_a}\mathrm{TF}_1(Z_0, \theta_k)\Pi^\top \mid \Pi\big]\big] &\text{(By (34), (35))}\\
=&\mathbb{E}_\Pi\big[\Pi\mathbb{E}[(R_{n+2} + \gamma\mathrm{TF}_1(Z'_0, \theta_k) - \mathrm{TF}_1(Z_0, \theta_k))\nabla_{Q_a}\mathrm{TF}_1(Z_0, \theta_k) \mid \Pi]\Pi^\top\big]\\
=&\mathbb{E}_\Pi\big[\Pi\mathbb{E}[(R_{n+2} + \gamma\mathrm{TF}_1(Z'_0, \theta_k) - \mathrm{TF}_1(Z_0, \theta_k))\nabla_{Q_a}\mathrm{TF}_1(Z_0, \theta_k)]\Pi^\top\big]\\
=&\mathbb{E}_\Pi\big[\Pi\mathrm{diag}(\Delta(Q_a))\Pi^\top\big]\\
=&\frac{1}{d}\mathrm{tr}(\Delta(Q_a))I_d &\text{(By Lemma A.2)}\\
=&\delta I_d.
\end{aligned}
$$

The proof is analogous for $\Delta(Q'_a) = \delta' I_d$ for some $\delta' \in \mathbb{R}$.

Suppose that $\Delta(p_{[2d+1]}) = \rho \in \mathbb{R}$, we now can conclude that

$$\Delta(\theta_k) = \left\{\Delta(P_0) = \begin{bmatrix} 0_{2d\times 2d} & 0_{2d\times 1}\\ 0_{1\times 2d} & \rho \end{bmatrix}, \Delta(Q_0) = \begin{bmatrix} \delta I_d & 0_{d\times d} & 0_{d\times 1}\\ \delta' I_d & 0_{d\times d} & 0_{d\times 1}\\ 0_{1\times d} & 0_{1\times d} & 0 \end{bmatrix}\right\}.$$

Therefore, according to (13), we get

$$
\begin{aligned}
&\theta_{k+1}\\
=&\theta_k + \alpha_k\Delta(\theta_k)\\
=&\left\{\begin{bmatrix} 0_{2d\times 2d} & 0_{2d\times 1}\\ 0_{1\times 2d} & \eta_k + \alpha_k\rho \end{bmatrix}, \begin{bmatrix} c_k + \alpha_k\delta I_d & 0_{d\times d} & 0_{d\times 1}\\ c'_k + \alpha_k\delta' I_d & 0_{d\times d} & 0_{d\times 1}\\ 0_{1\times d} & 0_{1\times d} & 0 \end{bmatrix}\right\} \in \Theta_*.
\end{aligned}
$$

$\square$

### A.4. Proof of Corollary 5.1

*Proof.* We recall from (5) that the embedding evolves according to

$$Z_{l+1} = Z_l + \frac{1}{n}P_l Z_l M(Z_l^\top Q_l Z_l).$$

We again refer to the elements in $Z_l$ as $\left\{(x_l^{(i)}, y_l^{(i)})\right\}_{i=1,\ldots,n+1}$ in the following way

$$Z_l = \begin{bmatrix} x_l^{(1)} & \cdots & x_l^{(n)} & x_l^{(n+1)}\\ y_l^{(1)} & \cdots & y_l^{(n)} & y_l^{(n+1)} \end{bmatrix},$$

where we recall that $Z_l \in \mathbb{R}^{(2d+1) \times (n+1)}, x_l^{(i)} \in \mathbb{R}^{2d}, y_l^{(i)} \in \mathbb{R}$. Sometimes, it is more convenient to refer to the first half and second half of $x_l^{(i)}$ separately, by, e.g., $\nu_l^{(i)} \in \mathbb{R}^d, \xi_l^{(i)} \in \mathbb{R}^d$, i.e., $x_l^{(i)} = \begin{bmatrix} \nu_l^{(i)} \\ \xi_l^{(i)} \end{bmatrix}$. Then, we have

$$
Z_l = \begin{bmatrix} \nu_l^{(1)} & \cdots & \nu_l^{(n)} & \nu_l^{(n+1)} \\ \xi_l^{(1)} & \cdots & \xi_l^{(n)} & \xi_l^{(n+1)} \\ y_l^{(1)} & \cdots & y_l^{(n)} & y_l^{(n+1)} \end{bmatrix}.
$$

We utilize the shorthands

$$
X_l = \begin{bmatrix} x_l^{(1)} & \cdots & x_l^{(n)} \end{bmatrix} \in \mathbb{R}^{2d \times n},
$$
$$
Y_l = \begin{bmatrix} y_l^{(1)} & \cdots & y_l^{(n)} \end{bmatrix} \in \mathbb{R}^{1 \times n}.
$$

Then we have

$$
Z_l = \begin{bmatrix} X_l & x_l^{(n+1)} \\ Y_l & y_l^{(n+1)} \end{bmatrix}.
$$

For the input $Z_0$, we assume $\xi_0^{(n+1)} = 0, y_0^{(n+1)} = 0$ but all other entries of $Z_0$ are arbitrary. We recall our definition of $M$ in (4) and $\{P_l^{\mathrm{RG}}, Q_l^{\mathrm{RG}}\}$ in (15). In particular, we can express $Q_l^{\mathrm{RG}}$ in a more compact way as

$$
M_1 \doteq \begin{bmatrix} -I_d & I_d \\ 0_{d \times d} & 0_{d \times d} \end{bmatrix} \in \mathbb{R}^{2d \times 2d},
$$
$$
M_2 \doteq -M_1
$$
$$
B_l \doteq \begin{bmatrix} C_l^\top & 0_{d \times d} \\ 0_{d \times d} & 0_{d \times d} \end{bmatrix} \in \mathbb{R}^{2d \times 2d},
$$
$$
A_l \doteq M_2^\top B_l M_1 = \begin{bmatrix} -C_l^\top & C_l^\top \\ C_l^\top & -C_l^\top \end{bmatrix} \in \mathbb{R}^{2d \times 2d},
$$
$$
Q_l^{\mathrm{RG}} \doteq \begin{bmatrix} A_l & 0_{2d \times 1} \\ 0_{1 \times 2d} & 0 \end{bmatrix} \in \mathbb{R}^{(2d+1) \times (2d+1)}.
$$

We then verify the following claims.

**Claim 1.** $X_l \equiv X_0, x_l^{(n+1)} \equiv x_0^{(n+1)}, \forall l.$

We note that $P_l^{\mathrm{RG}}$ is the key reason Claim 1 holds and is the same as the TD(0) case. Referring to A.1, we omit the proof of Claim 1 here.

**Claim 2.**

$$
Y_{l+1} = Y_l + \frac{1}{n} Y_l X^\top A_l X
$$
$$
y_{l+1}^{(n+1)} = y_l^{(n+1)} + \frac{1}{n} Y_l X^\top A_l x^{(n+1)}.
$$

Since the only difference between the true residual gradient and TD(0) configurations is the internal structure of $A_l$, we argue that it's irrelevant to Claim 2. We therefore again refer the readers to A.1 for a detailed proof.

**Claim 3.**

$$
y_{l+1}^{(i)} = y_0^{(i)} + \left\langle M_1 x^{(i)}, \frac{1}{n} \sum_{j=0}^{l} B_j^\top M_2 X Y_j^\top \right\rangle,
$$

for $i = 1, \ldots, n+1$.

By Claim 2, we can unroll $Y_{l+1}$ as

$$Y_{l+1} = Y_l + \frac{1}{n} Y_l X^\top A_l X$$

$$Y_l = Y_{l-1} + \frac{1}{n} Y_{l-1} X^\top A_{l-1} X$$

$$\vdots$$

$$Y_1 = Y_0 + \frac{1}{n} Y_0 X^\top A_0 X.$$

We can then compactly express $Y_{l+1}$ as

$$Y_{l+1} = Y_0 + \frac{1}{n} \sum_{j=0}^{l} Y_j X^\top A_j X.$$

Recall that we define $A_j = M_2^\top B_j M_1$. Then, we can rewrite $Y_{l+1}$ as

$$Y_{l+1} = Y_0 + \frac{1}{n} \sum_{j=0}^{l} Y_j X^\top M_2^\top B_j M_1 X.$$

With the identical procedure, we can easily rewrite $y_{l+1}^{(n+1)}$ as

$$y_{l+1}^{(n+1)} = y_0^{(n+1)} + \frac{1}{n} \sum_{j=0}^{l} Y_j X^\top M_2^\top B_j M_1 x^{(n+1)}.$$

In light of this, we define $\psi_0 \doteq 0$ and for $l = 0, \dots$

$$\psi_{l+1} \doteq \frac{1}{n} \sum_{j=0}^{l} B_j^\top M_2 X Y_j^\top \in \mathbb{R}^{2d}$$

$$= \psi_l + \frac{1}{n} B_l^\top M_2 X Y_l^\top \tag{36}$$

Then we can write

$$y_{l+1}^{(i)} = y_0^{(i)} + \left\langle M_1 x^{(i)}, \psi_{l+1} \right\rangle, \tag{37}$$

for $i = 1, \dots, n+1$, which is the claim we made. In particular, since we assume $y_0^{(n+1)} = 0$, we have

$$y_{l+1}^{(n+1)} = \left\langle M_1 x^{(n+1)}, \psi_{l+1} \right\rangle.$$

**Claim 4.** The bottom $d$ elements of $\psi_l$ are always 0, i.e., there exists a sequence $\{w_l \in \mathbb{R}^d\}$ such that we can express $\psi_l$ as

$$\psi_l = \begin{bmatrix} w_l \\ 0_{d \times 1} \end{bmatrix}.$$

for all $l = 0, 1, \dots, L$.

Since $B_l$ is the key reason Claim 4 holds and is identical to the TD(0) case, we refer the reader to A.1 for detailed proof.

Given all the claims above, we can then compute that

$$
\left\langle \psi_{l+1}, M_1 x^{(n+1)} \right\rangle
$$
$$
= \left\langle \psi_l, M_1 x^{(n+1)} \right\rangle + \frac{1}{n} \left\langle B_l^\top M_2 X Y_l^\top, M_1 x^{(n+1)} \right\rangle \tag{By (36)}
$$
$$
= \left\langle \psi_l, M_1 x^{(n+1)} \right\rangle + \frac{1}{n} \sum_{i=1}^n \left\langle B_l^\top M_2 x^{(i)} y_l^{(i)}, M_1 x^{(n+1)} \right\rangle
$$
$$
= \left\langle \psi_l, M_1 x^{(n+1)} \right\rangle + \frac{1}{n} \sum_{i=1}^n \left\langle B_l^\top M_2 x^{(i)} \left( \left\langle \psi_l, M_1 x^{(i)} \right\rangle + y_0^{(i)} \right), M_1 x^{(n+1)} \right\rangle \tag{By (37)}
$$
$$
= \left\langle \psi_l, M_1 x^{(n+1)} \right\rangle + \frac{1}{n} \sum_{i=1}^n \left\langle B_l^\top \begin{bmatrix} \nu^{(i)} - \xi^{(i)} \\ 0_{d \times 1} \end{bmatrix} \left( \left\langle \psi_l, \begin{bmatrix} -\nu^{(i)} + \xi^{(i)} \\ 0_{d \times 1} \end{bmatrix} \right\rangle + y_0^{(i)} \right), M_1 x^{(n+1)} \right\rangle
$$
$$
= \left\langle \psi_l, M_1 x^{(n+1)} \right\rangle + \frac{1}{n} \sum_{i=1}^n \left\langle \begin{bmatrix} C_l(\nu^{(i)} - \xi^{(i)}) \\ 0_{d \times 1} \end{bmatrix} \left( y_0^{(i)} + w_l^\top \xi^{(i)} - w_l^\top \nu^{(i)} \right), M_1 x^{(n+1)} \right\rangle \tag{By Claim 4}
$$
$$
= \left\langle \psi_l, M_1 x^{(n+1)} \right\rangle + \frac{1}{n} \sum_{i=1}^n \left\langle \begin{bmatrix} C_l(\nu^{(i)} - \xi^{(i)}) \left( y_0^{(i)} + w_l^\top \xi^{(i)} - w_l^\top \nu^{(i)} \right) \\ 0_{d \times 1} \end{bmatrix}, M_1 x^{(n+1)} \right\rangle
$$

This means

$$
\left\langle w_{l+1}, \nu^{(n+1)} \right\rangle = \left\langle w_l, \nu^{(n+1)} \right\rangle + \frac{1}{n} \sum_{i=1}^n \left\langle C_l \left( \nu^{(i)} - \xi^{(i)} \right) \left( y_0^{(i)} + w_l^\top \xi^{(i)} - w_l^\top \nu^{(i)} \right), \nu^{(n+1)} \right\rangle.
$$

Since the choice of the query $\nu^{(n+1)}$ is arbitrary, we get

$$
w_{l+1} = w_l + \frac{1}{n} \sum_{i=1}^n C_l \left( y_0^{(i)} + w_l^\top \xi^{(i)} - w_l^\top \nu^{(i)} \right) \left( \nu^{(i)} - \xi^{(i)} \right).
$$

In particular, when we construct $Z_0$ such that $\nu^{(i)} = \phi_{i-1}$, $\xi^{(i)} = \gamma \phi_i$ and $y_0^{(i)} = R_i$, we get

$$
w_{l+1} = w_l + \frac{1}{n} \sum_{i=1}^n C_l \left( R_i + \gamma w_l^\top \phi_i - w_l^\top \phi_{i-1} \right) \left( \phi_{i-1} - \gamma \phi_i \right)
$$

which is the update rule for pre-conditioned residual gradient learning. We also have

$$
y_l^{(n+1)} = \left\langle \psi_l, M_1 x^{(n+1)} \right\rangle = - \left\langle w_l, \phi^{(n+1)} \right\rangle.
$$

This concludes our proof. $\qquad\qquad\qquad\qquad\qquad\qquad\qquad\qquad\qquad\qquad\qquad\qquad\qquad\qquad\qquad\qquad$ □

### A.5. Proof of Corollary 5.2

*Proof.* The proof presented here closely mirrors the methodology and notation established in the proof of Theorem 3.1 from Appendix A.1. We begin by recalling the embedding evolution from (5) as,

$$
Z_{l+1} = Z_l + \frac{1}{n} P_l Z_l M^{\mathrm{TD}(\lambda)} (Z_l^\top Q_l Z_l).
$$

where we have substituted the original mask defined in (4) with the TD($\lambda$) mask in (17). We once again refer to the elements in $Z_l$ as $\left\{ (x_l^{(i)}, y_l^{(i)}) \right\}_{i=1,\dots,n+1}$ in the following way

$$
Z_l = \begin{bmatrix} x_l^{(1)} & \cdots & x_l^{(n)} & x_l^{(n+1)} \\ y_l^{(1)} & \cdots & y_l^{(n)} & y_l^{(n+1)} \end{bmatrix},
$$

where we recall that $Z_l \in \mathbb{R}^{(2d+1)\times(n+1)}$, $x_l^{(i)} \in \mathbb{R}^{2d}$, $y_l^{(i)} \in \mathbb{R}$. We utilize, $\nu_l^{(i)} \in \mathbb{R}^d$, $\xi_l^{(i)} \in \mathbb{R}^d$, to refer to the first half and second half of $x_l^{(i)}$ i.e., $x_l^{(i)} = \begin{bmatrix} \nu_l^{(i)} \\ \xi_l^{(i)} \end{bmatrix}$.

Then we have

$$
Z_l = \begin{bmatrix} \nu_l^{(1)} & \cdots & \nu_l^{(n)} & \nu_l^{(n+1)} \\ \xi_l^{(1)} & \cdots & \xi_l^{(n)} & \xi_l^{(n+1)} \\ y_l^{(1)} & \cdots & y_l^{(n)} & y_l^{(n+1)} \end{bmatrix}.
$$

We further define as shorthands,

$$
X_l = \begin{bmatrix} x_l^{(1)} & \cdots & x_l^{(n)} \end{bmatrix} \in \mathbb{R}^{2d\times n},
$$
$$
Y_l = \begin{bmatrix} y_l^{(1)} & \cdots & y_l^{(n)} \end{bmatrix} \in \mathbb{R}^{1\times n}.
$$

Then the blockwise structure of $Z_l$ can be succinctly expressed as:

$$
Z_l = \begin{bmatrix} X_l & x_l^{(n+1)} \\ Y_l & y_l^{(n+1)} \end{bmatrix}.
$$

We proceed to the formal arguments by paralleling those in Theorem 3.1. As in the theorem, we assume that certain initial conditions, such as $\xi_0^{(n+1)} = 0$ and $y_0^{(n+1)} = 0$, hold, but other entries of $Z_0$ are arbitrary. We recall our definition of $M^{\mathrm{TD}(\lambda)}$ in (17) and $\{P_l^{\mathrm{TD}}, Q_l^{\mathrm{TD}}\}_{l=0,\ldots,L-1}$ in (10). In particular, we can express $Q_l^{\mathrm{TD}}$ in a more compact way as

$$
M_1 \doteq \begin{bmatrix} -I_d & I_d \\ 0_{d\times d} & 0_{d\times d} \end{bmatrix} \in \mathbb{R}^{2d\times 2d},
$$
$$
B_l \doteq \begin{bmatrix} C_l^\top & 0_{d\times d} \\ 0_{d\times d} & 0_{d\times d} \end{bmatrix} \in \mathbb{R}^{2d\times 2d},
$$
$$
A_l \doteq B_l M_1 = \begin{bmatrix} -C_l^\top & C_l^\top \\ 0_{d\times d} & 0_{d\times d} \end{bmatrix} \in \mathbb{R}^{2d\times 2d},
$$
$$
Q_l^{\mathrm{TD}} \doteq \begin{bmatrix} A_l & 0_{2d\times 1} \\ 0_{1\times 2d} & 0 \end{bmatrix} \in \mathbb{R}^{(2d+1)\times(2d+1)},
$$

We now proceed with the following claims.

In subsequent steps, it sometimes is useful to refer to the matrix $M^{\mathrm{TD}(\lambda)}Z^\top$ in block form. Therefore, we will define $H^\top \in \mathbb{R}^{(n\times 2d)}$ as the first $n$ rows of $M_{\mathrm{TD}(\lambda)}Z^\top$ except for the last column, which we define as $Y_l^{(\lambda)} \in \mathbb{R}^n$.

$$
M^{\mathrm{TD}(\lambda)}Z_l^\top = \begin{bmatrix} H^\top & Y_l^{(\lambda)} \\ 0_{1\times 2d} & 0 \end{bmatrix} \in \mathbb{R}^{(n+1)\times(2d+1)}
$$

Let $h^{(i)}$ denote $i$-th column of $H$.

We proceed with the following claims.

**Claim 1.** $X_l \equiv X_0, x_l^{(n+1)} \equiv x_0^{(n+1)}, \forall l$.

Because we utilize the same definition of $P_l^{\mathrm{TD}}$ as in Theorem 3.1, the argument proving Claim 1 in Theorem 3.1 holds here as well. As a result, we drop all the subscripts of $X_l$, as well as subscripts of $x_l^{(i)}$ for $i = 1, \ldots, n+1$.

**Claim 2.** Let $H \in \mathbb{R}^{(2d\times n)}$, where the $i$-th column of $H$ is,

$$
h^{(i)} = \sum_{k=1}^{i} \lambda^{i-k} x^{(i)} \in \mathbb{R}^{2d}.
$$

Then we can write the updates for $Y_{l+1}$, and $y_{l+1}^{(n+1)}$ as,

$$Y_{l+1} = Y_l + \frac{1}{n} Y_l H^\top A_l X,$$

$$y_{l+1}^{(n+1)} = y_l^{(n+1)} + \frac{1}{n} Y_l H^\top A_l x^{(n+1)}.$$

We will show this by factoring the embedding evolution into the product of $P_l^{\text{TD}} Z_l$ and $M^{\text{TD}(\lambda)} Z_l^\top$, and $Q_l^{\text{TD}} Z_l$. Firstly, we have

$$P_l^{\text{TD}} Z_l = \begin{bmatrix} 0_{2d \times n} & 0_{2d \times 1} \\ Y_l & y_l^{(n+1)} \end{bmatrix}.$$

Next we analyze $M^{\text{TD}(\lambda)} Z_l^\top$. From basic matrix algebra we have,

$$M^{\text{TD}(\lambda)} Z^\top = \begin{bmatrix} 1 & 0 & 0 & 0 & \cdots & 0 & 0 \\ \lambda & 1 & 0 & 0 & \cdots & 0 & 0 \\ \lambda^2 & \lambda & 1 & 0 & \cdots & 0 & 0 \\ \lambda^3 & \lambda^2 & \lambda & 1 & \cdots & 0 & 0 \\ \vdots & \vdots & \vdots & \vdots & \ddots & \vdots & \vdots \\ \lambda^{n-1} & \lambda^{n-2} & \lambda^{n-3} & \lambda^{n-4} & \cdots & 1 & 0 \\ 0 & 0 & 0 & 0 & \cdots & 0 & 0 \end{bmatrix} \begin{bmatrix} x^{(1)\top} & y^{(1)} \\ x^{(2)\top} & y^{(2)} \\ x^{(3)\top} & y^{(3)} \\ \vdots & \vdots \\ x^{(n)\top} & y^{(n)} \\ x^{(n+1)\top} & 0 \end{bmatrix}$$

$$= \begin{bmatrix} x^{(1)\top} & y_l^{(1)} \\ x^{(2)\top} + \lambda x^{(1)\top} & y_l^{(2)} + \lambda y_l^{(2)} \\ \vdots & \vdots \\ \sum_{i=1}^n \lambda^{n-i} x_i^\top & \sum_{i=1}^n \lambda^{n-i} y_l^{(i)} \\ 0_{1 \times 2d} & 0 \end{bmatrix},$$

$$= \begin{bmatrix} h^{(1)\top} & y_l^{(1)} \\ h^{(2)\top} & y_l^{(2)} + \lambda y_l^{(1)} \\ \vdots & \vdots \\ h^{(n)\top} & \sum_{i=1}^n \lambda^{n-i} y_l^{(n)} \\ 0_{1 \times 2d} & 0 \end{bmatrix}$$

$$= \begin{bmatrix} H^\top & K_l^{(\lambda)} \\ 0_{1 \times 2d} & 0 \end{bmatrix},$$

where $K_l^{(\lambda)} \in \mathbb{R}^d$ is introduced for notation simplicity.

Then, we analyze $M^{\text{TD}(\lambda)} Z_l^\top Q_l^{\text{TD}} Z_l$. Applying the block matrix notations, we get

$$\left( M^{\text{TD}(\lambda)} Z_l^\top \right) Q_l^{\text{TD}} Z_l = \begin{bmatrix} H^\top & K_l^{(\lambda)} \\ 0_{1 \times 2d} & 0 \end{bmatrix} \begin{bmatrix} A_l & 0_{2d \times 1} \\ 0_{1 \times 2d} & 0 \end{bmatrix} \begin{bmatrix} X & x^{(n+1)} \\ Y_l & y_l^{(n+1)} \end{bmatrix}$$

$$= \begin{bmatrix} H^\top A_l & 0_{n \times 1} \\ 0_{1 \times 2d} & 0 \end{bmatrix} \begin{bmatrix} X & x^{(n+1)} \\ Y_l & y_l^{(n+1)} \end{bmatrix}$$

$$= \begin{bmatrix} H^\top A_l X & H^\top A_l x^{(n+1)} \\ 0_{1 \times 2d} & 0 \end{bmatrix}.$$

Combining the two, we get

$$P_l^{\text{TD}} Z_l \left( M^{\text{TD}(\lambda)} Z_l^\top Q_l^{\text{TD}} Z_l \right) = \begin{bmatrix} 0_{2d \times n} & 0_{2d \times 1} \\ Y_l & y_l^{(n+1)} \end{bmatrix} \begin{bmatrix} H^\top A_l X & H^\top A_l x^{(n+1)} \\ 0_{1 \times 2d} & 0 \end{bmatrix}$$

$$= \begin{bmatrix} 0_{2d \times n} & 0_{2d \times 1} \\ Y_l H^\top A_l X & Y_l H^\top A_l x^{(n+1)} \end{bmatrix}.$$

Hence, according to our update rule in (5), we get

$$Y_{l+1} = Y_l + \frac{1}{n} Y_l H^\top A_l X$$

$$y_{l+1}^{(n+1)} = y_l^{(n+1)} + \frac{1}{n} Y_l H^\top A_l x^{(n+1)}.$$

**Claim 3.**

$$y_{l+1}^{(i)} = y_0^{(i)} + \left\langle M_1 x^{(i)}, \frac{1}{n} \sum_{i=0}^{l} B_i^\top M_2 X Y_i^\top \right\rangle,$$

for $i = 1, \ldots, n+1$, where $M_2 = \begin{bmatrix} I_d & 0_{d \times d} \\ 0_{d \times d} & 0_{d \times d} \end{bmatrix}$.

Following Claim 2, we can unroll the recursive definition of $Y_{l+1}$ and express it compactly as,

$$Y_{l+1} = Y_0 + \frac{1}{n} \sum_{i=0}^{l} Y_i H^\top A_i X.$$

Recall that we define $A_i = B_i M_1$. Then, we can rewrite $Y_{l+1}$ as

$$Y_{l+1} = Y_0 + \frac{1}{n} \sum_{i=0}^{l} Y_i H^\top M_2 B_i M_1 X.$$

The introduction of $M_2$ here does not break the equivalence because $B_i = M_2 B_i$. However, it will help make our proof steps easier to comprehend later.

With the identical recursive unrolling procedure, we can rewrite $y_{l+1}^{(n+1)}$ as

$$y_{l+1}^{(n+1)} = y_0^{(n+1)} + \frac{1}{n} \sum_{i=0}^{l} Y_i H^\top M_2 B_i M_1 x^{(n+1)}.$$

In light of this, we define $\psi_0 \doteq 0$ and for $l = 0, \ldots$

$$\psi_{l+1} \doteq \frac{1}{n} \sum_{i=0}^{l} B_i^\top M_2 H Y_i^\top \in \mathbb{R}^{2d}. \tag{38}$$

Then we can write

$$y_{l+1}^{(i)} = y_0^{(i)} + \left\langle M_1 x^{(i)}, \psi_{l+1} \right\rangle, \tag{39}$$

for $i = 1, \ldots, n+1$, which is the claim we made. In particular, since we assume $y_0^{(n+1)} = 0$, we have

$$y_{l+1}^{(n+1)} = \left\langle M_1 x^{(n+1)}, \psi_{l+1} \right\rangle.$$

**Claim 4.** The bottom $d$ elements of $\psi_l$ are always 0, i.e., there exists a sequence $\{ w_l \in \mathbb{R}^d \}$ such that we can express $\psi_l$ as

$$\psi_l = \begin{bmatrix} w_l \\ 0_{d \times 1} \end{bmatrix}.$$

for all $l = 0, 1, \ldots, L$.

Because we utilize the same definition of $B_l$ as in Theorem 3.1 when defining $\psi_{l+1}$, the argument proving Claim 4 in Theorem 3.1 holds here as well. We omit the steps to avoid redundancy.

Given all the claims above, we can then compute that

$$
\left\langle \psi_{l+1}, M_1 x^{(n+1)} \right\rangle
$$
$$
= \left\langle \psi_l, M_1 x^{(n+1)} \right\rangle + \frac{1}{n} \left\langle B_l^\top M_2 H Y_l^\top, M_1 x^{(n+1)} \right\rangle \tag{By (38)}
$$
$$
= \left\langle \psi_l, M_1 x^{(n+1)} \right\rangle + \frac{1}{n} \sum_{i=1}^{n} \left\langle B_l^\top M_2 h^{(i)} y_l^{(i)}, M_1 x^{(n+1)} \right\rangle
$$
$$
= \left\langle \psi_l, M_1 x^{(n+1)} \right\rangle + \frac{1}{n} \sum_{i=1}^{n} \left\langle B_l^\top M_2 h^{(i)} \left( \left\langle \psi_l, M_1 x^{(i)} \right\rangle + y_0^{(i)} \right), M_1 x^{(n+1)} \right\rangle \tag{By (39)}
$$
$$
= \left\langle \psi_l, M_1 x^{(n+1)} \right\rangle + \frac{1}{n} \sum_{i=1}^{n} \left\langle B_l^\top \left[ \begin{array}{c} \left( \sum_{k=1}^{i} \lambda^{i-k} \nu^{(i)} \right) \\ 0_{d\times 1} \end{array} \right] \left( \left\langle \psi_l, \left[ \begin{array}{c} -\nu^{(i)} + \xi^{(i)} \\ 0_{d\times 1} \end{array} \right] \right\rangle + y_0^{(i)} \right), M_1 x^{(n+1)} \right\rangle
$$
$$
= \left\langle \psi_l, M_1 x^{(n+1)} \right\rangle + \frac{1}{n} \sum_{i=1}^{n} \left\langle \left[ \begin{array}{c} C_l \left( \sum_{k=1}^{i} \lambda^{i-k} \nu^{(i)} \right) \\ 0_{d\times 1} \end{array} \right] \left( y_0^{(i)} + w_l^\top \xi^{(i)} - w_l^\top \nu^{(i)} \right), M_1 x^{(n+1)} \right\rangle \tag{By Claim 4}
$$
$$
= \left\langle \psi_l, M_1 x^{(n+1)} \right\rangle + \frac{1}{n} \sum_{i=1}^{n} \left\langle \left[ \begin{array}{c} C_l \left( y_0^{(i)} + w_l^\top \xi^{(i)} - w_l^\top \nu^{(i)} \right) \left( \sum_{k=1}^{i} \lambda^{i-k} \nu^{(i)} \right) \\ 0_{d\times 1} \end{array} \right], M_1 x^{(n+1)} \right\rangle
$$

This means

$$
\left\langle w_{l+1}, \nu^{(n+1)} \right\rangle = \left\langle w_l, \nu^{(n+1)} \right\rangle + \frac{1}{n} \sum_{i=1}^{n} \left\langle C_l \left( y_0^{(i)} + w_l^\top \xi^{(i)} - w_l^\top \nu^{(i)} \right) \left( \sum_{k=1}^{i} \lambda^{i-k} \nu^{(i)} \right), \nu^{(n+1)} \right\rangle.
$$

Since the choice of the query $\nu^{(n+1)}$ is arbitrary, we get

$$
w_{l+1} = w_l + \frac{1}{n} \sum_{i=1}^{n} C_l \left( y_0^{(i)} + w_l^\top \xi^{(i)} - w_l^\top \nu^{(i)} \right) \left( \sum_{k=1}^{i} \lambda^{i-k} \nu^{(i)} \right).
$$

In particular, when we construct $Z_0$ such that $\nu^{(i)} = \phi_{i-1}$, $\xi^{(i)} = \gamma \phi_i$ and $y_0^{(i)} = R_i$, we get

$$
w_{l+1} = w_l + \frac{1}{n} \sum_{i=1}^{n} C_l \left( R_i + \gamma w_l^\top \phi_i - w_l^\top \phi_{i-1} \right) e_{i-1}
$$

where

$$
e_i = \sum_{k=1}^{i} \lambda^{i-k} \phi_k. \in \mathbb{R}^d
$$

which is the update rule for pre-conditioned TD($\lambda$). We also have

$$
y_l^{(n+1)} = \left\langle \psi_l, M_1 x^{(n+1)} \right\rangle = - \left\langle w_l, \phi^{(n+1)} \right\rangle.
$$

This concludes our proof. □

### A.6. Proof of Theorem 5.3

*Proof.* We recall from (21) that the embedding evolves according to

$$
Z_{l+1} = Z_l + \frac{1}{n} \text{TwoHead}(Z_l; P_l^{\overline{\text{TD}},(1)}, Q_l^{\overline{\text{TD}}}, M^{\overline{\text{TD}},(1)}, P_l^{\overline{\text{TD}},(2)}, Q_l^{\overline{\text{TD}}}, M^{\overline{\text{TD}},(2)}, W_l)
$$
$$
= Z_l + \frac{1}{n} W_l \left[ \begin{array}{c} \text{LinAttn}(Z_l; P_l^{\overline{\text{TD}},(1)}, Q_l^{\overline{\text{TD}}}, M^{\overline{\text{TD}},(1)}) \\ \text{LinAttn}(Z_l; P_l^{\overline{\text{TD}},(2)}, Q_l^{\overline{\text{TD}}}, M^{\overline{\text{TD}},(2)}) \end{array} \right]
$$

In this configuration, we refer to the elements in $Z_l$ as $\left\{(x_l^{(i)}, y_l^{(i)}, h_l^{(i)})\right\}_{i=1,\ldots,n+1}$ in the following way,

$$Z_l = \begin{bmatrix} x_l^{(1)} & \cdots & x_l^{(n)} & x_l^{(n+1)} \\ y_l^{(1)} & \cdots & y_l^{(n)} & y_l^{(n+1)} \\ h_l^{(1)} & \cdots & h_l^{(n)} & h_l^{(n+1)} \end{bmatrix},$$

where we recall that $Z_l \in \mathbb{R}^{(2d+2)\times(n+1)}, x_l^{(i)} \in \mathbb{R}^{2d}, y_l^{(i)} \in \mathbb{R}$ and $h_l^{(i)} \in \mathbb{R}$.

Sometimes, it is more convenient to refer to the first half and second half of $x_l^{(i)}$ separately, by, e.g., $\nu_l^{(i)} \in \mathbb{R}^d, \xi_l^{(i)} \in \mathbb{R}^d$, i.e., $x_l^{(i)} = \begin{bmatrix} \nu_l^{(i)} \\ \xi_l^{(i)} \end{bmatrix}$. Then we have

$$Z_l = \begin{bmatrix} \nu_l^{(1)} & \cdots & \nu_l^{(n)} & \nu_l^{(n+1)} \\ \xi_l^{(1)} & \cdots & \xi_l^{(n)} & \xi_l^{(n+1)} \\ y_l^{(1)} & \cdots & y_l^{(n)} & y_l^{(n+1)} \\ h_l^{(1)} & \cdots & h_l^{(n)} & h_l^{(n+1)} \end{bmatrix}.$$

We further define as shorthands

$$X_l \doteq \begin{bmatrix} x_l^{(1)} & \cdots & x_l^{(n)} \end{bmatrix} \in \mathbb{R}^{2d\times n},$$

$$Y_l \doteq \begin{bmatrix} y_l^{(1)} & \cdots & y_l^{(n)} \end{bmatrix} \in \mathbb{R}^{1\times n},$$

$$H_l \doteq \begin{bmatrix} h_l^{(1)} & \cdots & h_l^{(n)} \end{bmatrix} \in \mathbb{R}^{1\times n}.$$

Then we can express $Z_l$ as

$$Z_l = \begin{bmatrix} X_l & x_l^{(n+1)} \\ Y_l & y_l^{(n+1)} \\ H_l & h_l^{(n+1)} \end{bmatrix}.$$

For the input $Z_0$, we assume $\xi_0^{(n+1)} = 0$ and $h_0^{(i)} = 0$ for $i = 1, \ldots, n+1$. All other entries of $Z_0$ are arbitrary. We recall our definition of $M^{\overline{\text{TD}},(1)}, M^{\overline{\text{TD}},(2)}$ in (20), $\left\{ P_l^{\overline{\text{TD}},(1)}, P_l^{\overline{\text{TD}},(2)}, Q_l^{\overline{\text{TD}}}, W_l \right\}$ in (18) and (19). We again express $Q_l^{\overline{\text{TD}}}$ as

$$M_1 \doteq \begin{bmatrix} -I_d & I_d \\ 0_{d\times d} & 0_{d\times d} \end{bmatrix} \in \mathbb{R}^{2d\times 2d},$$

$$B_l \doteq \begin{bmatrix} C_l^\top & 0_{d\times d} \\ 0_{d\times d} & 0_{d\times d} \end{bmatrix} \in \mathbb{R}^{2d\times 2d},$$

$$A_l \doteq B_l M_1 = \begin{bmatrix} -C_l^\top & C_l^\top \\ 0_{d\times d} & 0_{d\times d} \end{bmatrix} \in \mathbb{R}^{2d\times 2d},$$

$$Q_l^{\overline{\text{TD}}} \doteq \begin{bmatrix} A_l & 0_{2d\times 2} \\ 0_{2\times 2d} & 0_{2\times 2} \end{bmatrix} \in \mathbb{R}^{(2d+2)\times(2d+2)}.$$

We now proceed with the following claims that assist in proving our main theorem.

**Claim 1.** $X_l \equiv X_0, x_l^{(n+1)} \equiv x_0^{(n+1)}, Y_l \equiv Y_0, y_l^{(n+1)} = y_0^{(n+1)}, \forall l.$

We define

$$V_l^{(1)} \doteq P_l^{\overline{\text{TD}},(1)} Z_l M^{\overline{\text{TD}},(1)} \left( Z_l^\top Q_l^{\overline{\text{TD}}} Z_l \right) \in \mathbb{R}^{(2d+2)\times(n+1)}$$

$$V_l^{(2)} \doteq P_l^{\overline{\text{TD}},(2)} Z_l M^{\overline{\text{TD}},(2)} \left( Z_l^\top Q_l^{\overline{\text{TD}}} Z_l \right) \in \mathbb{R}^{(2d+2)\times(n+1)}.$$

Then the evolution of the embedding can be written as

$$Z_{l+1} = Z_l + \frac{1}{n} W_l \begin{bmatrix} V_l^{(1)} \\ V_l^{(2)} \end{bmatrix}.$$

By simple matrix arithmetic, we realize $W_l$ is merely summing up the $(2d+1)$-th row of $V_l^{(1)}$ and the $(2d+2)$-th row of $V_l^{(2)}$ and putting the result on its bottom row. Thus, we have

$$W_l \begin{bmatrix} V_l^{(1)} \\ V_l^{(2)} \end{bmatrix} = \begin{bmatrix} 0_{(2d+1)\times(n+1)} \\ V_l^{(1)}(2d+1) + V_l^{(2)}(2d+2) \end{bmatrix} \in \mathbb{R}^{(2d+2)\times(n+1)},$$

where $V_l^{(1)}(2d+1)$ and $V_l^{(2)}(2d+2)$ respectively indicate the $(2d+1)$-th row of $V_l^{(1)}$ and the $(2d+2)$-th row of $V_l^{(2)}$. It clearly holds according to the update rule that

$$\begin{aligned} Z_{l+1}(1:2d+1) &= Z_l(1:2d+1) \\ \implies X_{l+1} &= X_l; \\ x_{l+1}^{(n+1)} &= x_l^{(n+1)}; \\ Y_{l+1} &= Y_l; \\ y_{l+1}^{(n+1)} &= y_l^{(n+1)}. \end{aligned}$$

Then, we can easily arrive at our claim by a simple induction. In light of this, we drop the subscripts of $X_l, x_l^{(i)}, Y_l$ and $y_l^{(i)}$ for all $i = 1, \ldots, n+1$ and write $Z_l$ as

$$Z_l = \begin{bmatrix} X & x^{(n+1)} \\ Y & y^{(n+1)} \\ H_l & h_l^{(n+1)} \end{bmatrix}.$$

**Claim 2.**

$$H_{l+1} = H_l + \frac{1}{n}(H_l + Y - \bar{Y})X^\top A_l X$$

$$h_{l+1}^{(n+1)} = h_l^{(n+1)} + \frac{1}{n}(H_l + Y - \bar{Y})X^\top A_l x^{(n+1)},$$

where $\bar{y}^{(i)} \doteq \sum_{k=1}^{i} \frac{y^{(k)}}{i}$ and $\bar{Y} \doteq \left[\bar{y}^{(1)}, \bar{y}^{(2)}, \ldots, \bar{y}^{(n)}\right] \in \mathbb{R}^{1\times n}$.

We show how this claim holds by investigating the function of each attention head in our formulation. The first attention head, corresponding to $V_l^{(1)}$ in claim 1, has the form

$$P_l^{\overline{\text{TD}},(1)} Z_l M^{\overline{\text{TD}},(1)} \left( Z_l^\top Q_l^{\overline{\text{TD}}} Z_l \right).$$

We first analyze $P_l^{\overline{\text{TD}},(1)} Z_l M^{\overline{\text{TD}},(1)}$. It should be clear that $P^{\overline{\text{TD}},(1)} Z_l$ selects out the $(2d+1)$-th row of $Z_l$ and gives us

$$P_l^{\overline{\text{TD}},(1)} = \begin{bmatrix} 0_{2d\times n} & 0_{2d\times 1} \\ Y & y^{(n+1)} \\ 0_{1\times n} & 0 \end{bmatrix}.$$

The matrix $M^{\overline{\text{TD}},(1)}$ is essentially computing $Y - \bar{Y}$ and filtering out the $(n+1)$-th entry when applied to $P_l^{\overline{\text{TD}},(1)} Z_l$. We

break down the steps here:

$$P_l^{\overline{\text{TD}},(1)} Z_l M^{\overline{\text{TD}},(1)}$$

$$=P_l^{\overline{\text{TD}},(1)} Z_l \big(I_{n+1} - U_{n+1}\text{diag}\big(\begin{bmatrix} 1 & \frac{1}{2} & \cdots & \frac{1}{n} \end{bmatrix}\big)\big) M^{\overline{\text{TD}},(2)}$$

$$=P_l^{\overline{\text{TD}},(1)} Z_l M^{\overline{\text{TD}},(2)} - P_l^{\overline{\text{TD}},(1)} Z_l U_{n+1}\text{diag}\big(\begin{bmatrix} 1 & \frac{1}{2} & \cdots & \frac{1}{n} \end{bmatrix}\big) M^{\overline{\text{TD}},(2)}$$

$$=\begin{bmatrix} 0_{2d\times n} & 0_{2d\times 1} \\ Y & 0 \\ 0_{1\times n} & 0 \end{bmatrix} - \begin{bmatrix} 0_{2d\times 1} & 0_{2d\times 1} & \cdots & 0_{2d\times 1} & 0_{2d\times 1} \\ y^{(1)} & \frac{1}{2}\big(y^{(1)}+y^{(2)}\big) & \cdots & \frac{1}{n}\sum_{i=1}^{n} y^{(i)} & \frac{1}{n+1}\sum_{i=1}^{n+1} y^{(i)} \\ 0 & 0 & \cdots & 0 & 0 \end{bmatrix} M^{\overline{\text{TD}},(2)}$$

$$=\begin{bmatrix} 0_{2d\times n} & 0_{2d\times 1} \\ Y & 0 \\ 0_{1\times n} & 0 \end{bmatrix} - \begin{bmatrix} 0_{2d\times n} & 0_{2d\times 1} \\ \bar{Y} & 0 \\ 0_{1\times n} & 0 \end{bmatrix}$$

$$=\begin{bmatrix} 0_{2d\times n} & 0_{2d\times 1} \\ Y - \bar{Y} & 0 \\ 0_{1\times n} & 0 \end{bmatrix}.$$

We then analyze the remaining product $Z_l^\top Q_l^{\overline{\text{TD}}} Z_l$.

$$Z_l^\top Q_l^{\overline{\text{TD}}} Z_l$$

$$=\begin{bmatrix} X^\top & Y^\top & H_l^\top \\ x^{(n+1)\top} & y^{(n+1)\top} & h_l^{(n+1)\top} \end{bmatrix} \begin{bmatrix} A_l & 0_{2d\times 1} & 0_{2d\times 1} \\ 0_{1\times 2d} & 0 & 0 \\ 0_{1\times 2d} & 0 & 0 \end{bmatrix} \begin{bmatrix} X & x^{(n+1)} \\ Y & y^{(n+1)} \\ H_l & h_l^{(n+1)} \end{bmatrix}$$

$$=\begin{bmatrix} X^\top A_l & 0_{n\times 1} & 0_{n\times 1} \\ x^{(n+1)\top} A_l & 0 & 0 \end{bmatrix} \begin{bmatrix} X & x^{(n+1)} \\ Y & y^{(n+1)} \\ H_l & h_l^{(n+1)} \end{bmatrix}$$

$$=\begin{bmatrix} X^\top A_l X & X^\top A_l x^{(n+1)} \\ x^{(n+1)\top} A_l X & x^{(n+1)\top} A_l x^{(n+1)} \end{bmatrix}.$$

Putting them together, we get

$$P_l^{\overline{\text{TD}},(1)} Z_l M^{\overline{\text{TD}},(1)}\big(Z_l^\top Q_l^{\overline{\text{TD}}} Z_l\big) = \begin{bmatrix} 0_{2d\times n} & 0_{2d\times 1} \\ Y - \bar{Y} & 0 \\ 0_{1\times n} & 0 \end{bmatrix} \begin{bmatrix} X^\top A_l X & X^\top A_l x^{(n+1)} \\ x^{(n+1)\top} A_l X & x^{(n+1)\top} A_l x^{(n+1)} \end{bmatrix}$$

$$= \begin{bmatrix} 0_{2d\times n} & 0_{2d\times 1} \\ (Y-\bar{Y})X^\top A_l X & (Y-\bar{Y})X^\top A_l x^{(n+1)} \\ 0_{1\times n} & 0 \end{bmatrix}.$$

The second attention head, corresponding to $V_l^{(2)}$ in claim 1, has the form

$$P_l^{\overline{\text{TD}},(2)} Z_l M^{\overline{\text{TD}},(2)}\big(Z_l^\top Q_l^{\overline{\text{TD}}} Z_l\big).$$

It's obvious that $P_l^{\overline{\text{TD}},(2)}$ selects out the $(2d+2)$-th row of $Z_l$ as

$$P_l^{\overline{\text{TD}},(2)} Z_l = \begin{bmatrix} 0_{(2d+1)\times n} & 0_{(2d+1)\times 1} \\ H_l & h_l^{(n+1)} \end{bmatrix}.$$

Applying the mask $M^{\overline{\text{TD}},(2)}$, we get

$$P_l^{\overline{\text{TD}},(2)} Z_l M^{\overline{\text{TD}},(2)} = \begin{bmatrix} 0_{(2d+1)\times n} & 0_{(2d+1)\times 1} \\ H_l & 0 \end{bmatrix}.$$

The product $Z_l^\top Q_l^{\overline{\text{TD}}} Z_l$ is identical to the first attention head. Hence, we see the computation of the second attention head gives us

$$P_l^{\overline{\text{TD}},(2)} Z_l M^{\overline{\text{TD}},(2)} \left( Z_l^\top Q_l^{\overline{\text{TD}}} Z_l \right)$$

$$= \begin{bmatrix} 0_{(2d+1)\times n} & 0_{(2d+1)\times 1} \\ H_l & 0 \end{bmatrix} \begin{bmatrix} X^\top A_l X & X^\top A_l x^{(n+1)} \\ x^{(n+1)\top} A_l X & x^{(n+1)\top} A_l x^{(n+1)} \end{bmatrix}$$

$$= \begin{bmatrix} 0_{(2d+1)\times n} & 0_{(2d+1)\times 1} \\ H_l X^\top A_l X & H_l X^\top A_l x^{(n+1)} \end{bmatrix}.$$

Lastly, the matrix $W_l$ combines the output from the two heads and gives us

$$W_l \begin{bmatrix} P_l^{\overline{\text{TD}},(1)} Z_l M^{\overline{\text{TD}},(1)} \left( Z_l^\top Q_l^{\overline{\text{TD}}} Z_l \right) \\ P_l^{\overline{\text{TD}},(2)} Z_l M^{\overline{\text{TD}},(2)} \left( Z_l^\top Q_l^{\overline{\text{TD}}} Z_l \right) \end{bmatrix} = \begin{bmatrix} 0_{(2d+1)\times n} & 0_{(2d+1)\times 1} \\ \left( H_l + Y - \bar{Y} \right) X^\top A_l X & \left( H_l + Y - \bar{Y} \right) X^\top A_l x^{(n+1)} \end{bmatrix}.$$

Hence, we obtain the update rule for $H_l$ and $h_l^{(n+1)}$ as

$$H_{l+1} = H_l + \frac{1}{n}(H_l + Y - \bar{Y}) X^\top A_l X$$

$$h_{l+1}^{(n+1)} = h_l^{(n+1)} + \frac{1}{n}(H_l + Y - \bar{Y}) X^\top A_l x^{(n+1)}$$

and claim 2 has been verified.

**Claim 3.**

$$h_{l+1}^{(i)} = \left\langle M_1 x^{(i)}, \frac{1}{n} \sum_{j=0}^{l} B_i^\top M_2 X (H_j + Y - \bar{Y})^\top \right\rangle,$$

for $i = 1, \ldots, n+1$, where $M_2 = \begin{bmatrix} I_d & 0_{d\times d} \\ 0_{d\times d} & 0_{d\times d} \end{bmatrix}$.

Following claim 2, we unroll $H_{l+1}$ as

$$H_{l+1} = H_l + \frac{1}{n}(H_l + Y - \bar{Y}) X^\top A_l X$$

$$H_l = H_{l-1} + \frac{1}{n}(H_{l-1} + Y - \bar{Y}) X^\top A_{l-1} X$$

$$\vdots$$

$$H_1 = H_0 + \frac{1}{n}(H_0 + Y - \bar{Y}) X^\top A_0 X.$$

We therefore can express $H_{l+1}$ as

$$H_{l+1} = H_0 + \frac{1}{n} \sum_{j=0}^{l} (H_j + Y - \bar{Y}) X^\top A_j X.$$

Recall that we have defined $A_j \doteq B_j M_1$ and assumed $H_0 = 0$. Then, we have

$$H_{l+1} = \frac{1}{n} \sum_{j=0}^{l} (H_j + Y - \bar{Y}) X^\top M_2 B_j M_1 X.$$

Note that the introduction of $M_2$ here does not break the equivalence because $B_j = M_2 B_j$. We include it in our expression for the convenience of the main proof later.

With the identical procedure, we can easily rewrite $h_{l+1}^{(n+1)}$ as

$$h_{l+1}^{(n+1)} = \frac{1}{n} \sum_{j=0}^{l} (H_j + Y - \bar{Y}) X^\top M_2 B_j M_1 x^{(n+1)}.$$

In light of this, we define $\psi_0 \doteq 0$, and for $l = 0, \dots$

$$\psi_{l+1} = \frac{1}{n} \sum_{j=0}^{l} B_j^\top M_2 X (H_j + Y - \bar{Y})^\top \in \mathbb{R}^{2d}.$$

We then can write

$$h_{l+1}^{(i)} = \left\langle M_1 x^{(i)}, \psi_{l+1} \right\rangle \tag{40}$$

for $i = 1, \dots, n+1$, which is the claim we made.

**Claim 4.** The bottom $d$ elements of $\psi_l$ are always 0, i.e., there exists a sequence $\{w_l \in \mathbb{R}^d\}$ such that we can express $\psi_l$ as

$$\psi_l = \begin{bmatrix} w_l \\ 0_{d \times 1} \end{bmatrix}.$$

for all $l = 0, 1, \dots, L$.

Since our $B_j$ here is identical to the proof of Theorem 3.1 in A.1 for $j = 0, 1, \dots$, Claim 4 holds for the same reason. We therefore omit the proof details to avoid repetition.

Given all the claims above, we proceed to prove our main theorem.

$$\left\langle \psi_{l+1}, M_1 x^{(n+1)} \right\rangle$$
$$= \left\langle \psi_l, M_1 x^{(n+1)} \right\rangle + \frac{1}{n} \left\langle B_l^\top M_2 X (H_l + Y - \bar{Y})^\top, M_1 x^{(n+1)} \right\rangle$$
$$= \left\langle \psi_l, M_1 x^{(n+1)} \right\rangle + \frac{1}{n} \sum_{i=1}^{n} \left\langle B_l^\top M_2 x^{(i)} (h_l^{(i)} + y^{(i)} - \bar{y}^{(i)}), M_1 x^{(n+1)} \right\rangle$$
$$= \left\langle \psi_l, M_1 x^{(n+1)} \right\rangle + \frac{1}{n} \sum_{i=1}^{n} \left\langle B_l^\top M_2 x^{(i)} \left( \left\langle \psi_l, M_1 x^{(i)} \right\rangle + y^{(i)} - \bar{y}^{(i)} \right), M_1 x^{(n+1)} \right\rangle \qquad \text{(By (40))}$$
$$= \left\langle \psi_l, M_1 x^{(n+1)} \right\rangle + \frac{1}{n} \sum_{i=1}^{n} \left\langle B_l^\top \begin{bmatrix} \nu^{(i)} \\ 0_{d \times 1} \end{bmatrix} \left( \left\langle \psi_l, \begin{bmatrix} -\nu^{(i)} + \xi^{(i)} \\ 0_{d \times 1} \end{bmatrix} \right\rangle + y^{(i)} - \bar{y}^{(i)} \right), M_1 x^{(n+1)} \right\rangle$$
$$= \left\langle \psi_l, M_1 x^{(n+1)} \right\rangle + \frac{1}{n} \sum_{i=1}^{n} \left\langle \begin{bmatrix} C_l \nu^{(i)} \\ 0_{d \times 1} \end{bmatrix} \left( y^{(i)} - \bar{y}^{(i)} + w_l^\top \xi^{(i)} - w_l^\top \nu^{(i)} \right), M_1 x^{(n+1)} \right\rangle \qquad \text{(By Claim 4)}$$
$$= \left\langle \psi_l, M_1 x^{(n+1)} \right\rangle + \frac{1}{n} \sum_{i=1}^{n} \left\langle \begin{bmatrix} C_l \nu^{(i)} \left( y^{(i)} - \bar{y}^{(i)} + w_l^\top \xi^{(i)} - w_l^\top \nu^{(i)} \right) \\ 0_{d \times 1} \end{bmatrix}, M_1 x^{(n+1)} \right\rangle$$

This means

$$\left\langle w_{l+1}, \nu^{(n+1)} \right\rangle = \left\langle w_l, \nu^{(n+1)} \right\rangle + \frac{1}{n} \sum_{i=1}^{n} \left\langle C_l \nu^{(i)} \left( y^{(i)} - \bar{y}^{(i)} + w_l^\top \xi^{(i)} - w_l^\top \nu^{(i)} \right), \nu^{(n+1)} \right\rangle.$$

Since the choice of the query $\nu^{(n+1)}$ is arbitrary, we get

$$w_{l+1} = w_l + \frac{1}{n} \sum_{i=1}^{n} C_l \left( y^{(i)} - \bar{y}^{(i)} + w_l^\top \xi^{(i)} - w_l^\top \nu^{(i)} \right) \nu^{(i)}.$$

In particular, when we construct $Z_0$ such that $\nu^{(i)} = \phi_{i-1}, \xi^{(i)} = \phi_i$ and $y^{(i)} = R_i$, we get

$$w_{l+1} = w_l + \frac{1}{n} \sum_{i=1}^{n} C_l \big( R_i - \bar{r}_i + w_l^\top \phi_i - w_l^\top \phi_{i-1} \big) \phi_{i-1}$$

which is the update rule for pre-conditioned average reward TD learning. We also have

$$h_l^{(n+1)} = \left\langle \psi_l, M_1 x^{(n+1)} \right\rangle = -\left\langle w_l, \phi^{(n+1)} \right\rangle.$$

This concludes our proof. $\qquad\qquad\qquad\qquad\qquad\qquad\qquad\qquad\qquad\qquad\qquad\qquad\quad$ $\square$

## B. Evaluation Task Generation

To generate the evaluation tasks used to meta-train our transformer in Algorithm 1, we utilize Boyan's chain, detailed in Figure 2. Notably, we make some minor adjustments to the original Boyan's chain in Boyan (1999) to make it an infinite horizon chain.

Recall that an evaluation task is defined by the tuple $(p_0, p, r, \phi)$. We consider Boyan's chain MRPs with $m$ states. To construct $p_0$, we first sample a $m$-dimensional random vector uniformly in $[0,1]^m$ and then normalize it to a probability distribution. To construct $p$, we keep the structure of Boyan's chain but randomize the transition probabilities. In particular, the transition function $p$ can be regarded as a random matrix taking value in $\mathbb{R}^{m \times m}$. For simplifying presentation, we use both $p(s, s')$ and $p(s'|s)$ to denote probability of transitioning to $s'$ from $s$. In particular, for $i = 1, \dots, m-2$, we set $p(i, i+1) = \epsilon$ and $p(i, i+2) = 1-\epsilon$, with $\epsilon$ sampled uniformly from $(0, 1)$. For the last two states, we have $p(m|m-1) = 1$ and $p(\cdot|m)$ is a random distribution over all states. Each element of the vector $r \in \mathbb{R}^m$ and the matrix $\phi \in \mathbb{R}^{d \times m}$ are sampled i.i.d. from a uniform distribution over $[-1, 1]$. The overall task generation process is summarized in Algorithm 2. Almost surely, no task will be generated twice. In our experiments in the main text, we use Boyan Chain MRPs which consist of $m = 10$ states each with feature dimension $d = 4$.

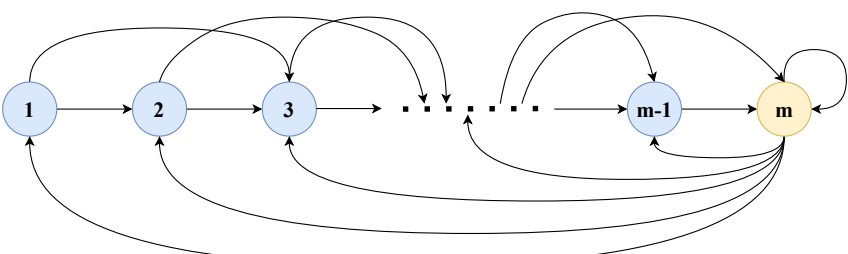

Figure 2: Boyan's Chain of $m$ States

**Representable Value Function.** With the above sampling procedure, there is no guarantee that the true value function $v$ is always representable by the features. In other words, there is no guarantee that there exists a $w \in \mathbb{R}^d$ satisfying $v(s) = \langle w, \phi(s) \rangle$ for all $s \in \mathcal{S}$. Most of our experiments use this setup. It is, however, also beneficial sometimes to work with evaluation tasks where the true value function is guaranteed to be representable. Algorithm 3 achieves this by randomly generating a $w_*$ first and compute $v(s) \doteq \langle w_*, \phi(s) \rangle$. The reward is then analytically computed as $r \doteq (I_m - \gamma p)v$. We recall that in the above we regard $p$ as a matrix in $\mathbb{R}^{m \times m}$.

---

**Algorithm 2** Boyan Chain MRP and Feature Generation (Non-Representable)

---

1: **Input:** state space size $m = |\mathcal{S}|$, feature dimension $d$
2: **for** $s \in \mathcal{S}$ **do**
3:     $\phi(s) \sim \text{Uniform}\left[(-1, 1)^d\right]$   // feature
4: **end for**
5: $p_0 \sim \text{Uniform}\left[(0, 1)^m\right]$   // initial distribution
6: $p_0 \leftarrow p_0 / \sum_s p_0(s)$
7: $r \sim \text{Uniform}\left[(-1, 1)^m\right]$   // reward function
8: $p \leftarrow 0_{m \times m}$   // transition function
9: **for** $i = 1, \ldots, m - 2$ **do**
10:     $\epsilon \sim \text{Uniform}\left[(0, 1)\right]$
11:     $p(i, i + 1) \leftarrow \epsilon$
12:     $p(i, i + 2) \leftarrow 1 - \epsilon$
13: **end for**
14: $p(m - 1, m) \leftarrow 1$
15: $z \leftarrow \text{Uniform}\left[(0, 1)^m\right]$
16: $z \leftarrow z / \sum_s z(s)$
17: $p(m, 1 : m) \leftarrow z$
18: **Output:** MRP $(p_0, p, r)$ and feature map $\phi$

---

## C. Additional Experiments with Linear Transformers

### C.1. Experiment Setup

We use Algorithm 2 as $d_{\text{task}}$ for the experiments in the main text with Boyan's chain of 10 states. In particular, we consider a context of length $n = 30$, feature dimension $d = 4$, and utilize a discount factor $\gamma = 0.9$. In Section 4, we consider a 3-layer transformer ($L = 3$), but additional analyses on the sensitivity to the number of transformer layers ($L$) and results from a larger scale experiment with $d = 8, n = 60$, and $|\mathcal{S}| = 20$ are presented in C.2. We also explore non-autoregressive (i.e., "sequential") layer configurations in C.3.

When training our transformer, we utilize an Adam optimizer (Kingma & Ba, 2015) with an initial learning rate of $\alpha = 0.001$, and weight decay rate of $1 \times 10^{-6}$. $P_0$ and $Q_0$ are randomly initialized using Xavier initialization with a gain of $0.1$. We trained our transformer on $k = 4000$ different evaluation tasks. For each task, we generated a trajectory of length $\tau = 347$, resulting in $\tau - n - 2 = 320$ transformer parameter updates.

Since the models in these experiments are small ($\sim 10$ KB), we did not use any GPU's during our experiments. We trained our transformers on a standard Intel i9-12900-HK CPU and training each transformer took $\sim 20$ minutes.

For implementation[3], we used NumPy (Harris et al., 2020) to process the data and construct Boyan's chain, PyTorch (Ansel et al., 2024) to define and train our models, and Matplotlib (Hunter, 2007) plus SciencePlots (Garrett, 2021) to generate our figures.

#### C.1.1. TRAINED TRANSFORMER ELEMENT-WISE CONVERGENCE METRICS

To visualize the parameters of the linear transformer trained by Algorithm 1, we report element-wise metrics. For $P_0$, we report the value of its bottom-right entry, which, as noted in (10), should approach one if the transformer is learning to implement TD. The other entries of $P_0$ should remain close to zero. Additionally, we report the average absolute value of the elements of $P_0$, excluding the bottom-right entry, to check if these elements stay near zero during training.

For $Q_0$, we recall from (10) that if the transformer learned to implement normal batch TD, the upper-left $d \times d$ block of the matrix should converge to some $-I_d$, while the upper-right $d \times d$ block (excluding the last column) should converge to $I_d$. To visualize this, we report the trace of the upper-left $d \times d$ block, and the trace of the upper-right $d \times d$ block (excluding the last column). The rest of the elements of $Q_0$ should remain close to 0, and to verify this, we report the average absolute value of the entries of $Q_0$, excluding the entries that were utilized in computing the traces.

---

[3]The code will be made publicly available upon publication.

---

**Algorithm 3** Boyan Chain MRP and Feature Generation (Representable)

---

1: **Input:** state space size $m = |\mathcal{S}|$, feature dimension $d$, discount factor $\gamma$
2: $w^* \sim \text{Uniform}\left[(-1, 1)^d\right]$  // ground-truth weight
3: **for** $s \in \mathcal{S}$ **do**
4:     $\phi(s) \sim \text{Uniform}\left[(-1, 1)^d\right]$  // feature
5:     $v(s) \leftarrow \langle w^*, \phi(s) \rangle$  // ground-truth value function
6: **end for**
7: $p_0 \sim \text{Uniform}\left[(0, 1)^m\right]$  // initial distribution
8: $p_0 \leftarrow p_0 / \sum_s p_0(s)$
9: $p \leftarrow 0_{m \times m}$  // transition function
10: **for** $i = 1, \ldots, m - 2$ **do**
11:     $\epsilon \sim \text{Uniform}\left[(0, 1)\right]$
12:     $p(i, i+1) \leftarrow \epsilon$
13:     $p(i, i+2) \leftarrow 1 - \epsilon$
14: **end for**
15: $p(m-1, m) \leftarrow 1$
16: $z \leftarrow \text{Uniform}\left[(0, 1)^m\right]$
17: $z \leftarrow z / \sum_s z(s)$
18: $p(m, 1:m) \leftarrow z$
19: $r \leftarrow (I_m - \gamma p) v$  // reward function
20: **Output:** MRP $(p_0, p, r)$ and feature map $\phi$

---

Since, $P_0$ and $Q_0$ are in the same product in (3) we sometimes observe during training that $P_0$ converges to $-P_0^{\text{TD}}$ and $Q_0$ converges to $-Q_0^{\text{TD}}$ simultaneously. When visualizing the matrices, we negate both $P_0$ and $Q_0$ when this occurs.

It's also worth noting that in Theorem 3.1 we prove a $L$-layer transformer parameterized as in (10) with $C_0 = I_d$ implements $L$ steps of batch TD exactly with a fixed update rate of one. However, the transformer trained using Algorithm 1 could learn to perform TD with an arbitrary learning rate ($\alpha$ in (8)). Therefore, even if the final trained $P_0$ and $Q_0$ differ from their constructions in (10) by some scaling factor, the resulting algorithm implemented by the trained transformer will still be implementing TD. In light of this, we rescale $P_0$ and $Q_0$ before visualization. In particular, we divide $P_0$ and $Q_0$ by the maximum of the absolute values of their entries respectively, such that they both stay in the range $[-1, 1]$ after rescaling.

### C.1.2. Trained Transformer and Batch TD Comparison Metrics

To compare the transformers with batch TD we report several metrics following von Oswald et al. (2023); Akyürek et al. (2023). Given a context $C \in \mathbb{R}^{(2d+1) \times n}$ and a query $\phi \in \mathbb{R}^d$, we construct the prompt as

$$
Z^{(\phi, C)} \doteq \left[ C \quad \begin{bmatrix} \phi \\ 0_{d \times 1} \\ 0 \end{bmatrix} \right].
$$

We will suppress the context $C$ in subscript when it does not confuse. We use $Z^{(s)} \doteq Z^{(\phi(s))}$ as shorthand. We use $d_p$ to denote the stationary distribution of the MRP with transition function $p$ and assume the context $C$ is constructed based on trajectories sampled from this MRP. Then, we can define $v_\theta \in \mathbb{R}^{|\mathcal{S}|}$, where $v_\theta(s) \doteq \text{TF}_L(Z_0^{(s)}; \theta)$ for each $s \in \mathcal{S}$. Notably, $v_\theta$ is then the value function estimation induced by the transformer parameterized by $\theta \doteq \{(P_l, Q_l)\}$ given the context $C$. In the rest of the appendix, we will use $\theta_{\text{TF}}$ as the learned parameter from Algorithm 1. As a result, $v_{\text{TF}} \doteq v_{\theta_{\text{TF}}}$ denotes the learned value function.

We define $\theta_{\text{TD}} \doteq \left\{ (P_l^{\text{TD}}, Q_l^{\text{TD}}) \right\}_{l=0,\ldots,L-1}$ with $C_l = \alpha I$ (see (10)) and

$$
v_{\text{TD}}(s) \doteq \text{TF}_L(Z_0^{(s)}; \theta_{\text{TD}}).
$$

In light of Theorem 3.1, $v_{\text{TD}}$ is then the value function estimation obtained by running the batch TD algorithm (11) on the context $C$ for $L$ iterations, using a constant learning rate $\alpha$.

We would like to compare the two functions $v_{\text{TF}}$ and $v_{\text{TD}}$ to future examine the behavior of the learned transformers. However, $v_{\text{TD}}$ is not well-defined yet because it still has a free parameter $\alpha$, the learning rate. (von Oswald et al., 2023) resolve a similar issue in the in-context regression setting via using a line search to find the (empirically) optimal $\alpha$. Inspired by (von Oswald et al., 2023), we also aim to find the empirically optimal $\alpha$ for $v_{\text{TD}}$. We recall that $v_{\text{TD}}$ is essentially the transformer $\text{TF}_L(Z_0^{(s)}; \theta_{\text{TD}})$ with only 1 single free parameter $\alpha$. We then train this transformer with Algorithm 1. We observe that $\alpha$ quickly converges and use the converged $\alpha$ to complete the definition of $v_{\text{TD}}$. We are now ready to present different metrics to compare $v_{\text{TF}}$ and $v_{\text{TD}}$. We recall that both are dependent on the context $C$.

**Value Difference (VD).** First for a given context $C$, we compute the Value Difference (VD) to measure the difference between the value function approximated by the trained transformer and the value function learned by batch TD, weighted by the stationary distribution. To this end, we define,

$$\text{VD}(v_{\text{TF}}, v_{\text{TD}}) \doteq \|v_{\text{TF}} - v_{\text{TD}}\|_{d_p}^2,$$

We recall that $d_p \in \mathbb{R}^{|\mathcal{S}|}$ is the stationary distribution of the MRP and the weighted $\ell_2$ norm is defined as $\|v\|_d \doteq \sqrt{\sum_s v(s)^2 d(s)}$.

**Implicit Weight Similarity (IWS).** We recall that $v_{\text{TD}}$ is a linear function, i.e., $v_{\text{TD}}(s) = \langle w_L, \phi(s) \rangle$ with $w_L$ defined in Theorem 3.1. We refer to this $w_L$ as $w_{\text{TD}}$ for clarity. The learned value function $v_{\text{TF}}$ is, however, not linear even when estimated by a linear transformer. Following Akyürek et al. (2023), we compute the best linear approximation of $v_{\text{TF}}$. In particular, given a context $C$, we define

$$w_{\text{TF}} \doteq \arg \min_w \|\Phi w - v_{\text{TF}}\|_{d_p}.$$

Here $\Phi \in \mathbb{R}^{|\mathcal{S}| \times d}$ is the feature matrix, each of which is $\phi(s)^\top$. Such a $w_{\text{TF}}$ is referred to as implicit weight in Akyürek et al. (2023). Following Akyürek et al. (2023), we define

$$\text{IWS}(v_{\text{TF}}, v_{\text{TD}}) \doteq d_{\cos}(w_{\text{TF}}, w_{\text{TD}})$$

to measure the similarity between $w_{\text{TF}}$ and $w_{\text{TD}}$. Here $d_{\cos}(\cdot, \cdot)$ computes the cos similarity between two vectors.

**Sensitivity Similarity (SS).** Recall that $v_{\text{TF}}(s) = \text{TF}_L(Z_0^{(s)}; \theta_{\text{TF}})$ and $v_{\text{TD}}(s) = \text{TF}_L(Z_0^{(s)}; \theta_{\text{TD}})$. In other words, given a context $C$, both $v_{\text{TF}}(s)$ and $v_{\text{TD}}(s)$ are functions of $\phi(s)$. Following von Oswald et al. (2023), we then measure the sensitivity of $v_{\text{TF}}(s)$ and $v_{\text{TD}}(s)$ w.r.t. $\phi(s)$. This similarity is easily captured by gradients. In particular, we define

$$\text{SS}(v_{\text{TF}}, v_{\text{TD}}) \doteq \sum_s d_p(s) d_{\cos} \left( \nabla_\phi \text{TF}_L(Z_0^{(\phi)}; \theta_{\text{TF}}) \Big|_{\phi=\phi(s)}, \nabla_\phi \text{TF}_L(Z_0^{(\phi)}; \theta_{\text{TD}}) \Big|_{\phi=\phi(s)} \right).$$

Notably, it trivially holds that

$$w_{\text{TD}} = \nabla_\phi \text{TF}_L(Z_0^{(\phi)}; \theta_{\text{TD}}) \Big|_{\phi=\phi(s)}.$$

We note that the element-wise converge of learned transformer parameters (e.g., Figure 1a) is the most definite evidence for the emergence of in-context TD. The three metrics defined in this section are only auxiliary when linear attention is concerned. That being said, **the three metrics are important when nonlinear attention is concerned**.

### C.2. Autoregressive Linear Transformers with $L = 1, 2, 3, 4$ Layers

In this section, we present the experimental results for autoregressive linear transformers with different numbers of layers. In Figure 3, we present the element-wise convergence metrics for autoregressive transformers with $L = 1, 2, 4$ layers. The plot with $L = 3$ is in Figure 1 in the main text. We can see that for the $L = 1$ case, $P_0$ and $Q_0$ converge to the construction in Corollary 3.2, which, as proved, implements TD(0) in the single layer case. For the $L = 2, 4$ cases, we see that $P_0$ and $Q_0$ converge to the construction in Theorem 3.1. We also observe that as the number of transformer layers $L$ increases, the learned parameters are more aligned with the construction of $P_0^{\text{TD}}$ and $Q_0^{\text{TD}}$ with $C_0 = I$.

We also present the comparison of the learned transformer with batch TD according to the metrics described in Appendix C.1.2. In Figure 4, we present the value difference, implicit weight similarity, and sensitivity similarity. In Figures 4a – 4d,

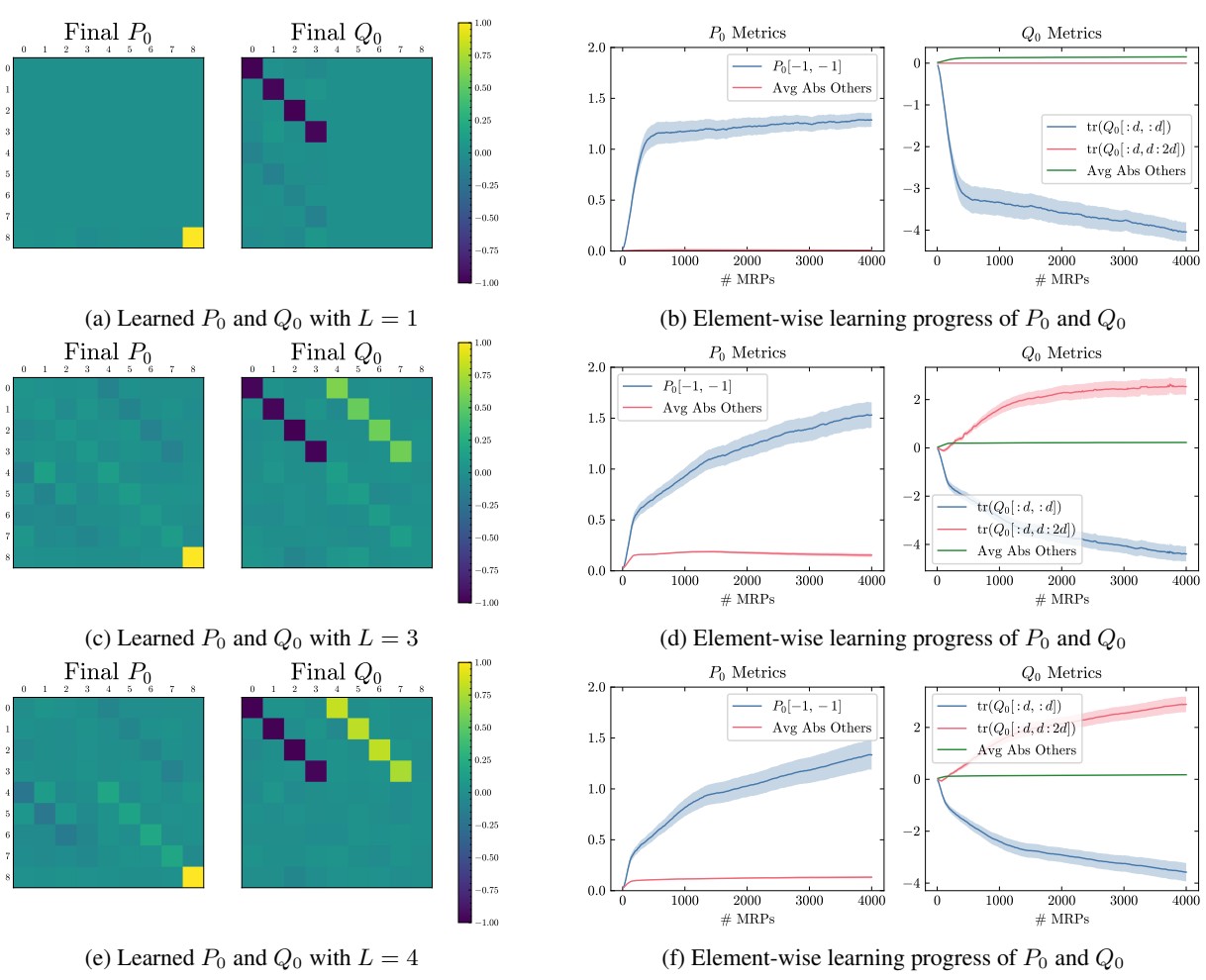

Figure 3: Visualization of the learned **autoregressive** transformers and the learning progress. Averaged across 30 seeds and the shaded region denotes the standard errors. See Appendix C.1.1 for details about normalization of $P_0$ and $Q_0$ before visualization.

we present the results for different transformer layer numbers $L = 1, 2, 3, 4$. In Figure 4e, we present the metrics for a 3-layer transformer, but we increase the feature dimension to $d = 8$ and also the context length to $n = 60$.

In all instances, we see strong similarity between the trained linear transformers and batch TD. We see that the cosine similarities of the sensitivities are near one, as are the implicit weight similarities. Additionally, the value difference approaches zero during training. This further demonstrates that the autoregressive linear transformers trained according to Algorithm 1 learn to implement TD(0).

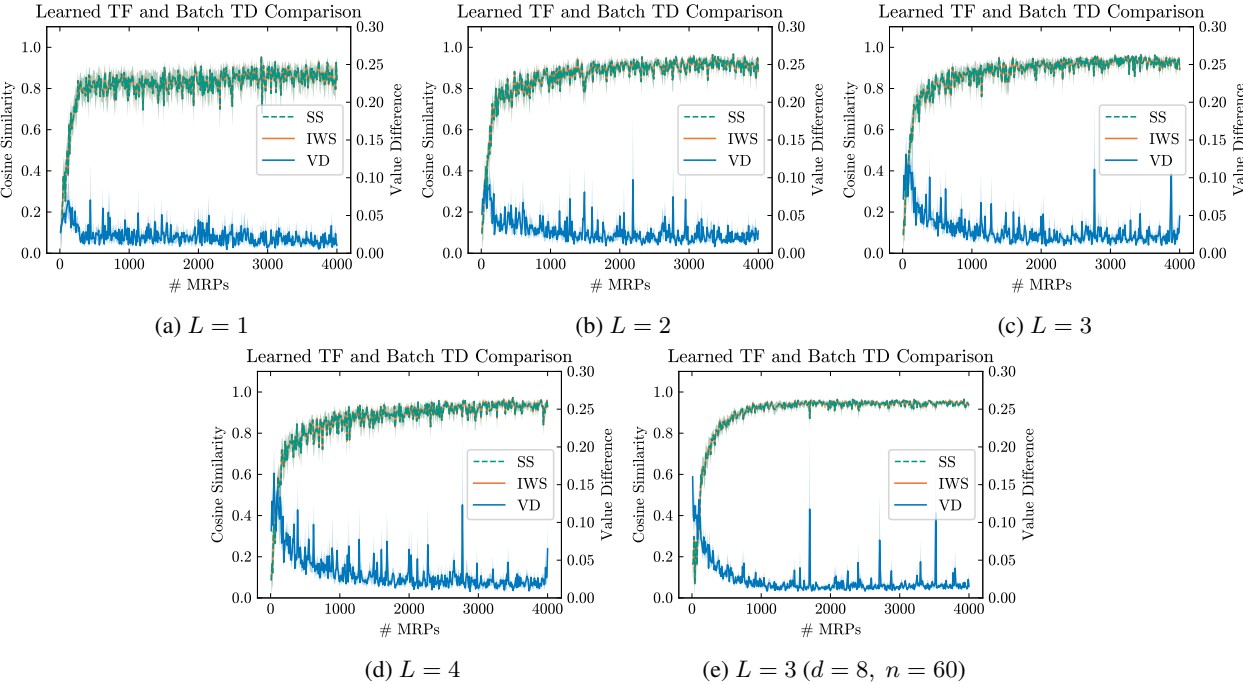

Figure 4: Value difference (VD), implicit weight similarity (IWS), and sensitivity similarity (SS) between the learned **autoregressive** transformers and batch TD with different layers. All curves are averaged over 30 seeds and the shaded regions are the standard errors.

### C.3. Sequential Transformers with $L = 2, 3, 4$ Layers

So far, we have been using linear transformers with one parametric attention layer applied repeatedly for $L$ steps to implement an $L$-layer transformer. Another natural architecture in contrast with the autoregressive transformer is a sequential transformer with $L$ distinct attention layers, where the embedding passes over each layer exactly once during one pass of forward propagation.

In this section, we repeat the same experiments we conduct on the autoregressive transformer with sequential transformers with $L = 2, 3, 4$ as their architectures coincide when $L = 1$. We compare the sequential transformers with batch TD(0) and report the three metrics in Figure 5. We observe that the implicit weight similarity and the sensitivity similarity grow drastically to near 1, and the value difference drops considerably after a few hundred MRPs for all three layer numbers. It suggests that sequential transformers trained via Algorithm 1 are functionally close to batch TD.

Figure 6 shows the visualization of the converged $\{P_l, Q_l\}_{l=0,1,2}$ of a 3-layer sequential linear transformer and their element-wise convergence. Sequential transformers exhibit very special patterns in their learned weights. We see that the input layer converges to a pattern very close to our configuration in Theorem (3.1). However, the deeper the layer, we observe the more the diagonal of $Q_l[1 : d, d + 1 : 2d]$ fades. The $P$ matrices, on the other hand, follow our configuration closely, especially for the final layer. We speculate this pattern emerges because sequential transformers have more parametric attention layers and thus can assign a slightly different role to each layer but together implement batch TD(0) as suggested by the black-box functional comparison in Figure 5.

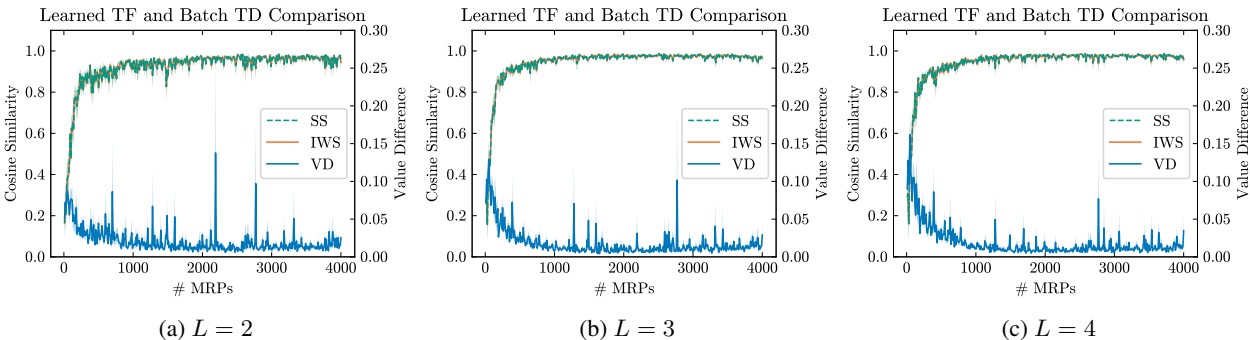

(a) $L = 2$        (b) $L = 3$        (c) $L = 4$

Figure 5: Value difference (VD), implicit weight similarity (IWS), and sensitivity similarity (SS) between the learned **autoregressive** transformers and batch TD with different layers. All curves are averaged over 30 seeds and the shaded regions are the standard errors.

(a) Learned $P_0$ and $Q_0$            (b) Element-wise learning progress of $P_0$ and $Q_0$

(c) Learned $P_1$ and $Q_1$            (d) Element-wise learning progress of $P_1$ and $Q_1$

(e) Learned $P_2$ and $Q_2$            (f) Element-wise learning progress of $P_2$ and $Q_2$

Figure 6: Visualization of the learned $L = 3$ **sequential** transformers and the learning progress. Averaged across 30 seeds and the shaded region denotes the standard errors. See Appendix C.1.1 for details about normalization of $P_0$ and $Q_0$ before visualization.

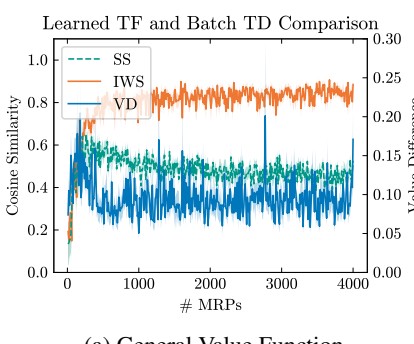

(a) General Value Function

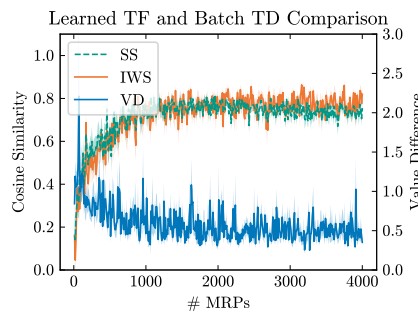

(b) Representable Value Function

Figure 7: Value difference (VD), implicit weight similarity (IWS), and sensitivity similarity (SS) between the learned softmax transformers and linear batch TD. All curves are averaged over 30 seeds and the shaded regions are the standard errors.

## D. Nonlinear Attention

Until now, we have focused on only linear attention. In this section, we empirically investigate original transformers with the softmax function. Given a matrix $Z$, we recall that self-attention computes it embedding as

$$\text{Attn}(Z; P, Q) = PZM\text{softmax}\big(Z^\top QZ\big).$$

Let $Z_l \in \mathbb{R}^{(2d+1)\times(n+1)}$ denote the input to the $l$-th layer, the output of an $L$-layer transformer with parameters $\{(P_l, Q_l)\}_{l=0,\dots,L-1}$ is then computed as

$$Z_{l+1} = Z_l + \tfrac{1}{n}\text{Attn}(Z_l; P_l, Q_l) = Z_l + \tfrac{1}{n}PZM\text{softmax}\big(Z^\top QZ\big).$$

Analogous to the linear transformer, we define

$$\widetilde{\text{TF}}_L\Big(Z_0; \{P_l, Q_l\}_{l=0,1\dots,L-1}\Big) \doteq -Z_L[2d+1, n+1].$$

As a shorthand, we use $\widetilde{\text{TF}}_L(Z_0)$ to denote the output of the softmax transformers given prompt $Z_0$. We use the same training procedure (Algorithm 1) to train the softmax transformers. In particular, we consider a 3-layer autoregressive softmax transformer.

Notably, the three metrics in Appendix C.1.2 apply to softmax transformers as well. We still compare the learned softmax transformer with the linear batch TD in (11). In other words, the $v_{\text{TD}}$ related quantities are the same, and we only recompute $v_{\text{TF}}$ related quantities in Appendix C.1.2. As shown in Figure 7a, the value difference remains small and the implicit weight similarity increases. This suggests that the learned softmax transformer behaves similarly to linear batch TD. The sensitivity similarity, however, drops. This is expected. The learned softmax transformer $\widetilde{\text{TF}}_L$ is unlikely to be a linear function w.r.t. to the query while $v_{\text{TD}}$ is linear w.r.t. the query. So their gradients w.r.t. the query are unlikely to match. To further investigate this hypothesis, we additionally consider evaluation tasks where the true value function is guaranteed to be representable (Algorithm 3) and is thus a linear function w.r.t. the state feature. This provides more incentives for the learned softmax transformer to behave like a linear function. As shown in Figure 7b, the sensitivity similarity now increases.

## E. Numerical Verification of Proofs

We provide numerical verification for our proofs by construction (Theorem 3.1, Corollary 5.1, Corollary 5.2, and Theorem 5.3) as a sanity check. In particular, we plot $\log\big|-\langle\phi_n, w_l\rangle - y_l^{n+1}\big|$ against the number of layers $l$. For example, for Theorem 3.1, we first randomly generate $Z_0$ and $\{C_l\}$. Then $y_l^{(n+1)}$ is computed by unrolling the transformer layer by layer following (5) while $w_l$ is computed iteration by iteration following (11). We use double-precision floats and run for 30 seeds, each with a new prompt. As shown in Figure 8, even after 40 layers / iterations, the difference is still in the order of $10^{-10}$. It is not strictly 0 because of numerical errors. It sometimes increases because of the accumulation of numerical errors.

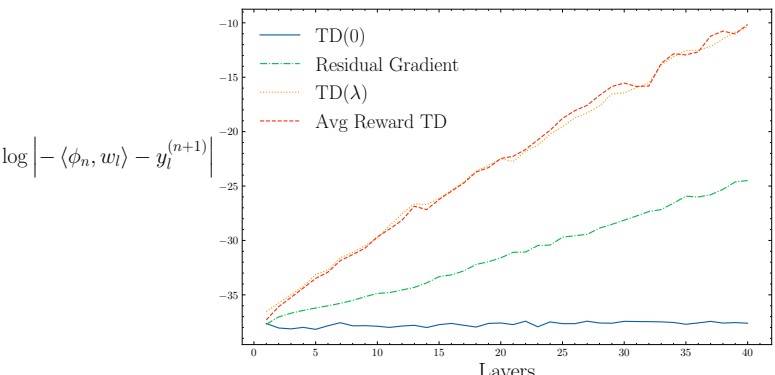

Figure 8: Differences between transformer output and batch TD output. Curves are averaged over 30 random seeds with the (invisible) shaded region showing the standard errors.

