# OpenReview forum: "Transformers Learn Temporal Difference Methods for In-Context  Reinforcement Learning"
_ICML.cc/2024/Workshop/ICL — ICML 2024 Workshop ICL Contributedtalk_

### Official Review · Reviewer_fYuW · 2024-06-07
**Review for Transformers Learn Temporal Difference Methods for In-Context Reinforcement Learning**

**Rating:** 2
**Fit:** 3
**Confidence:** 2

**Workshop Review:**

This paper shows that theoretically and empirically transformer models can implement temporal difference methods through in-context learning. It is highly relevant to the workshop and general community, as it demonstrates another aspect in which in-context learning can be applied to the reinforcement learning problem. The paper is also written clearly and has several theoretical results to support the paper's main thesis.

**Reason For Not Giving Higher Score:**

The empirical evaluations are very simple at the moment, despite the ability to scale up to harder RL problems with the power of transformer models.

**Reason For Not Giving Lower Score:**

This paper provides new understanding about how transformer models can model TD methods.

---

### Official Review · Reviewer_LRDS · 2024-06-12
**Great paper on in-context / memory-based meta reinforcement learning implementing TD algorithms**

**Rating:** 3
**Fit:** 3
**Confidence:** 2

**Workshop Review:**

I really enjoyed reading this paper about how Transformers can implement TD algorithms. It is well written and easy to follow (although I haven't rigorously checked the proofs) and I think a timely paper investigating in-context learning [also worth mentioning earlier references, eg 1] not just in supervised learning but also in RL.

While, as the authors rightly point out, in-context RL has been explored extensively in the past, how the implemented strategies relate to known RL algorithms is less clear. While not too surprising that Transformers can implement TD as universal function approximators, the authors do provide a theoretical construction and empirical evidence that with the right data distribution those models can be practically trained to do so. In the future I would love to see how this can be scaled up to discover novel TD-based RL algorithms in-context, related to previous attempts, e.g. [2,3,4].

There are a few typing errors in the paper e.g. line 99 right and 71 left.

[1] Hochreiter, Sepp, A. Steven Younger, and Peter R. Conwell. "Learning to learn using gradient descent." Artificial Neural Networks—ICANN 2001: International Conference Vienna, Austria, August 21–25, 2001 Proceedings 11. Springer Berlin Heidelberg, 2001.
[2] Kirsch, Louis, et al. "Introducing symmetries to black box meta reinforcement learning." Proceedings of the AAAI Conference on Artificial Intelligence. Vol. 36. No. 7. 2022.
[3] Lu, Chris, et al. "Structured state space models for in-context reinforcement learning." Advances in Neural Information Processing Systems 36 (2024).
[4] Kirsch, Louis, et al. "Towards general-purpose in-context learning agents." Workshop on Distribution Shifts, 37th Conference on Neural Information Processing Systems (NeurIPS 2023), OpenReview. net, 2023.

**Reason For Not Giving Higher Score:**

N/A

**Reason For Not Giving Lower Score:**

I could see this paper as a talk at the workshop! While I am not too surprised about the result, and I would love to see larger empirical studies, I think it is a good stepping stone towards understanding in-context RL and scaling it up to search for novel (TD-based) RL algorithms.

---

### Meta-Review · Area_Chair_sNuy · 2024-06-14

**Recommendation:** 3

**Metareview:**

The present paper presents a construction of transformer weights which implement temporal difference learning td(0). Through experiments they demonstrate that transformers really do learn these constructions during training.

Both reviewers point out the relevance of the submission to the ICL workshop, with reviewer LRDS pointing out that it's a timely submission and well-written. I encourage the authors to add the missing citations to their work.

I agree with reviewer LRDS on the quality of the submission and recommend acceptance as a talk.

---

### Decision · Program_Chairs · 2024-06-17

Accept (Contributed talk)